# Nanomaterials for Photocatalytic Inactivation and Eradication of *Candida* spp. Biofilms in Healthcare Environment: A Novel Approach in Modern Clinical Practice

**DOI:** 10.3390/molecules30234500

**Published:** 2025-11-21

**Authors:** Karolina Kraus, Paweł Mikziński, Jarosław Widelski, Emil Paluch

**Affiliations:** 1Faculty of Medicine, Wroclaw Medical University, Wyb. Pasteura 1, 50-376 Wroclaw, Poland; karolina.kraus@student.umw.edu.pl (K.K.); pawel.mikzinski@student.umw.edu.pl (P.M.); 2Department of Pharmacognosy with Medicinal Plants Garden, Lublin Medical University, 20-093 Lublin, Poland; jaroslaw.widelski@umlub.pl; 3Department of Microbiology, Faculty of Medicine, Wroclaw Medical University, Tytusa Chalubinskiego 4, 50-376 Wroclaw, Poland

**Keywords:** nanomaterials, nanoparticles, carbon dots, nanoenzymes, photocatalysis, photocatalytic inactivation, biofilm formation, *Candida* spp. biofilms

## Abstract

Modern medicine is facing a significant challenge in dealing with infections caused by *Candida* spp. and the biofilms they form. Although there are numerous treatment methods available for *Candida* species, standard therapeutic protocols are increasingly failing, particularly in cases of chronic local infections, such as those affecting immunocompromised patients (e.g., due to immunosuppression or diabetes). In such cases, a promising approach is to use nanomaterials to inactivate and eradicate *Candida* spp. and their biofilms. In order to limit the spread of *Candida* spp. and their biofilms within the healthcare environment, thereby reducing the risk of patient infection, photocatalysis appears to be a noteworthy method for improving therapeutic outcomes. *Candida* spp. biofilms are difficult to eradicate because they possess multiple resistance mechanisms—including protective extracellular matrix, efflux pumps, quorum sensing, persister and Goliath cells—which collectively enhance drug tolerance, adhesion, and survival rates under antifungal treatment. The use of nanomaterials, such as nanoparticles, carbon dots, or nanozymes for photocatalytic processes, seems to be a promising solution, showing outstanding results in *Candida* spp. biofilm disruption and inactivation. This is due to their superior biofilm penetration, effective destruction of proteins and enzymes, destabilization of EPS, degradation of nucleic acids, and reduced drug resistance. We collected the most important nanomaterials useful in combating *Candida* spp. biofilm and organized the photocatalysis mechanism of action in its disruption. Based on current research, we have compiled modern strategies involving nanomaterials and their photocatalytic activity for potential application in the healthcare environment, with the aim of reducing the presence of *Candida* spp. biofilms and, consequently, lowering the incidence of *Candida* spp.-related infections.

## 1. Biofilm *Candida* spp. as a Healthcare Problem

*Candida* spp. remain a major challenge in contemporary healthcare, even in advanced tertiary and specialized centers. Five species—*Candida albicans*, *C. glabrata*, *C. parapsilosis*, *C. tropicalis*, and *C. krusei*—are responsible for the majority of clinically relevant infections. Fungal pathogens are estimated to account for approximately 15% of healthcare-associated infections, with *Candida* species causing 70–90% of invasive cases. Invasive candidiasis is associated with high mortality, which can exceed 70% [1]. *Candida* spp. are estimated to cause approximately 700,000 cases of invasive candidiasis globally each year [2]. Candidaemia constitutes the predominant clinical manifestation of invasive candidiasis, with the majority of episodes arising in the context of healthcare-associated infections [3]. Invasive candidiasis and candidaemia mainly occur in immunocompromised or critically ill patients, whereas mucocutaneous forms may develop in otherwise healthy individuals. Increasing antifungal resistance and novel host-related risk factors contribute to difficult-to-treat infections, driving higher healthcare costs, morbidity, and mortality. The emergence of *Candida auris* and fluconazole-resistant *Candida parapsilosis* represents a major global threat, particularly in hospitals, due to ease of transmission and limited efficacy of standard antifungal agents [4]. Nowadays, *Candida auris* is a major concern in critical care, with candidiasis as the leading fungal infection, contributing—along with other fungi—to an estimated 13 million cases and 1.5 million deaths annually worldwide [5].

With advancements in medical science, an increasing number of patients are receiving effective long-term treatment for various types of cancer and autoimmune diseases. This progress, however, correlates with a rise in the population of immunocompromised individuals within healthcare settings [6]. Additionally, prolonged and inappropriate pharmacotherapy fosters the emergence of highly resistant hospital-associated microbial populations, including *Candida* spp. and their biofilms. Although antifungal agents such as azoles and echinocandins are widely used to treat *Candida* infections, mortality rates remain elevated, reaching up to 45% [7].

*Candida* biofilms are capable of establishing on diverse substrates, encompassing biological tissues such as mucosal surfaces, organs, and blood vessels, as well as abiotic surfaces. In particular, medical devices that come into contact with patients—including stents, shunts, implants, endotracheal tubes, various catheters, and prosthetic joints—serve as frequent sites for biofilm formation [8].

## 2. Biofilm Formation by *Candida* spp.

### 2.1. Biofilm Formation Problem

As mentioned above, in recent years the topic of bacterial and fungal biofilms has gained considerable importance in various medical fields due to their increasing clinical relevance. A biofilm, as a complex structure, formed by multi-species communities of microorganisms, provides the microorganisms with numerous advantages, enabling them to survive under unfavorable conditions and making them significantly more difficult to eradicate compared to their planktonic (non-biofilm) forms [9].

In the case of *Candida* spp., biofilm development is a complex, multi-stage process regulated by various factors originating from both the yeast and the host. The formation of biofilms by *Candida* species can occur within 38–72 h. Host factors that promote *Candida* biofilm formation include immune suppression, exposure of adhesion-promoting surfaces (medical implants), altered physiological conditions (e.g., pH, nutrients, oxygen), and disruption of the normal microbiota [10]. The process of biofilm formation is generally divided into the following stages: adhesion, proliferation, maturation, and dispersion.

### 2.2. Adhesion

The first stage of biofilm development is adhesion, during which individual fungal cells attach to a suitable surface. High adhesion capacity to various surfaces, including human skin and mucosal tissues, is one of the key characteristics of *Candida* spp. In fungi, adhesion is mediated by cell wall glycoproteins (adhesins) that facilitate interactions with surfaces, other cells, and host tissues, and their expression varies depending on the growth phase [11].

The adhesion process has been most extensively studied in *C. albicans*. Among the most thoroughly characterized surface adhesins are proteins encoded by the ALS (agglutinin-like sequence) gene family (*Als1–9*), together with *Eap1* and *Hwp1*. During the initial attachment, *C. albicans* cells can roll across the surface, which exposes amyloid regions of adhesins and promotes stable binding.

Adherence of biofilm-forming *C. albicans* cells differs between the initial attachment phase and the proliferation/maturation phase of biofilm formation. A study by McCall et al. [12] highlights the most crucial adhesion proteins throughout biofilm development. The adhesin-coding genes *Eap1*, *Hwp2*, *Hyr1*, and Ihd1 appear to be particularly critical for mediating initial attachment. Additionally, during the early stage of yeast cell attachment to surfaces, Als1p and Als5p contribute to this process through their amyloid-forming domains, which are essential for both cell–cell aggregation and cell–surface adhesion. Adhesion is essential for successful biofilm growth and maturation. It has been suggested that hyphal formation in *C. albicans* enhances the maintenance of adhesion, thereby promoting further biofilm development. A hypha-specific adhesin, Hyr1p, belonging to the Hyr family, plays a key role in this process. Additionally, the transcriptional regulators Efg1 and Bcr1 are critical at this stage, as they control the expression of multiple adhesin genes. Several GPI-anchored proteins, including Pga1p, Pga10p, Ecm33p, Mp65p, Msb2p, Pbr1p, Arc18p, Pmt1p, Mnn9p, Spt7p, orf19.831, and Ihd1p, also contribute to adhesion during biofilm maturation.

In the later stages of biofilm growth, the gene coding Ywp1 (yeast wall protein 1), as well as other genes mentioned later, play a significant role [13,14].

Similarly to *C. albicans*, adhesion of *C. glabrata* to host tissues and abiotic surfaces is mediated by a large set of cell wall adhesins, many of which are GPI-anchored proteins. These adhesins, including the lectin-like Epa family, display modular structures with N-terminal binding domains and tandem repeat regions that contribute to binding specificity and strength. The diversity of adhesins in *C. glabrata* allows it to attach to a wide range of host ligands, resist phagocyte-mediated clearance, and support dissemination, highlighting the importance of adhesion in its pathogenicity [15].

In addition to *C. albicans* and *C. glabrata*, adhesion of *C. tropicalis* is also mediated by a variety of cell wall adhesins. Similarly to *C. albicans*, *C. tropicalis* expresses members of the ALS family, which play a key role in binding to host tissues and abiotic surfaces. While detailed studies in *C. tropicalis* are still limited, several *ALS* genes (e.g., *CtrALS1–3*, *CtrALS5*, *CtrALS1028*) have been identified, showing differential expression depending on cell morphology, growth conditions, and whether cells are planktonic or sessile. Differences in the N-terminal adhesion domains and central tandem repeats contribute to variation in adhesion strength and specificity. Additionally, *C. tropicalis* expresses Hwp1 and proteins homologous to the Iff/Hyr family, which facilitate attachment to epithelial and ECM components, similarly to *C. albicans*. These observations suggest that although the repertoire of adhesins in *C. tropicalis* is less studied, it employs molecular mechanisms of adhesion closely resembling those of *C. albicans*, highlighting conserved strategies among pathogenic *Candida* species for colonization and biofilm formation [16].

### 2.3. Proliferation and Maturing

This stage is characterized by the growth and expansion of cells within the developing biofilm. Concurrently, fungal cells secrete extracellular polymeric substances (EPS), typically composed of glycoproteins, polysaccharides, lipids, acids, and other biomolecules. The progressive accumulation of EPS results in the establishment of a three-dimensional extracellular matrix. In *Candida* spp., the matrix composition varies across species. According to Zarnowski et al. [17], the extracellular matrix (ECM) of biofilms formed by the most prevalent species, *C. albicans*, consists predominantly of proteins and glycoproteins (≈55% by weight), followed by carbohydrates (≈25%), lipids (≈15%), and extracellular DNA (≈5%). This matrix, built from both self-produced EPS and external components, including host-derived elements such as erythrocytes, epithelial cells, urothelial cells, and neutrophils, forms a complex and heterogeneous barrier that provides protection and multiple advantages to fungal cells, such as drug resistance. At the end of the maturation process, a three-dimensional multilayered biofilm is formed, consisting of yeast, pseudohyphal, and hyphal cells embedded within an ECM [18,19]. Typically, *Candida* biofilms consist of a surface-attached layer of yeast cells overlain by an interwoven network of hyphae and pseudohyphae, all enclosed within ECM. The development of this structure is controlled by various transcription factors, such as Efg1p (driving the yeast-to-hypha transition), Tec1p, and Rlm1p (involved in ECM production) [14].

Despite some similarities, mature biofilm architecture differs among *Candida* species. *C. albicans* forms thick, hypha-rich layers, *C. glabrata* produces thinner yeast-dominated films, *C. tropicalis* develops dense communities, while *C. parapsilosis* tends to create loosely organized structures. These variations affect ECM composition, antifungal resistance, and pathogenicity [20].

### 2.4. Dispersal Phase

The mature biofilm eventually enters the dispersal phase, during which cells detach and colonize new surfaces. This stage is central to the persistence and recurrence of *Candida* infections, as released cells can establish growth at distant sites and escape unfavorable local conditions. Wall et al. [21] demonstrated that dispersal is not restricted to mature biofilms but may also occur early in development, with the growth phase showing the highest release rate. Environmental parameters, including nutrient availability and pH, further modulate this process.

Genetic regulation is equally important: studies by Uppuluri et al. [22] identified PES1 as essential for dispersal, while overexpression of *PES1*, *NRG1*, and several *SAP* genes strongly promotes cell release. Moreover, modulation of yeast wall protein 1 (Ywp1), known for its antiadhesive activity, appears to be a key factor in dispersion control [23]. All phases of *Candida* spp. formation, along with the genes involved, are illustrated in Figure 1.

## 3. *Candida* spp. Biofilm Resistance Factors

### 3.1. Extracellular Matrix

As mentioned earlier, *Candida* spp. biofilms derive much of their drug resistance from the three-dimensional structure of the extracellular matrix. Composed mainly of proteins, polysaccharides, lipids, and extracellular DNA, it forms both a physical barrier that limits drug penetration and a structural scaffold for the biofilm. Polysaccharides, like glucans and mannans, within the matrix can bind and sequester antifungal agents, significantly increasing biofilm tolerance compared with planktonic cells. Glucan synthesis, mediated by Fks1, enhances this effect by boosting the matrix’s drug-binding capacity, preventing drugs from reaching their cellular targets. This resistance mechanism is biofilm-specific, as altering FKS1 expression in planktonic cells does not affect drug susceptibility. Similar matrix-mediated resistance has been observed in biofilms of other *Candida* species, and polymicrobial biofilms further modify the matrix composition and its ability to reduce drug efficacy [24].

### 3.2. Efflux Pumps

Efflux pumps are membrane transport proteins that actively export antimicrobial agents from cells, contributing to drug resistance. They can be specific or multidrug transporters and are found in many organisms, including fungi. By reducing intracellular drug concentrations, efflux pumps enable fungi to survive exposure to structurally unrelated antibiotics, leading to multidrug resistance (MDR) and characteristic tolerance to toxic compounds [25].

This method of drug resistance can be observed in *Candida* species. In planktonic cells, exposure to antifungal agents can trigger upregulation of efflux pump expression, whereas in biofilms, efflux pumps are typically upregulated from the early stages of adhesion and remain active throughout the entire biofilm development process [26].

The upregulation of efflux pump expression has been particularly observed in *C. albicans* biofilms; however, studies have also shown that the situation is similar in *C. tropicalis* and *C. glabrata* biofilms [27,28].

### 3.3. Quorum Sensing

Quorum sensing is a crucial mechanism in *Candida* biofilms, coordinating intercellular communication via signaling molecules such as farnesol and tyrosol. This process enables the biofilm to regulate its growth and morphology in response to nutritional and environmental conditions, contributing to the biofilm’s resistance potential. Specifically, farnesol facilitates the transition from hyphal to yeast forms, whereas tyrosol induces the opposite shift from yeast to hyphae [29].

### 3.4. Persister Cells

Another resistance mechanism associated with the biofilm structure is the presence of persister cells. Identified in *Candida* biofilms, these cells exhibit multidrug tolerance and can enter a metabolically inactive state that enables them to withstand high concentrations of antifungal agents. Unlike the majority of susceptible cells, persisters can survive under otherwise lethal antifungal conditions. This small subpopulation plays a critical role in biofilm resilience, as it allows regrowth and repopulation after the initial killing of the more vulnerable fungal cells [30].

### 3.5. Goliath Cells

Among the mechanisms underlying *Candida* albicans biofilm resistance, unique biofilm-forming traits have been attributed to the so-called Goliath cells. These unusually enlarged cells (approximately four times the size of standard yeast cells) exhibit markedly enhanced adhesive properties. Their formation is induced by zinc deprivation, leading to a phenotype characterized by increased chitin exposure, delayed hyphal development, and hyperadhesion.

Malavia et al. [31] demonstrated that this phenotype is regulated by the transcription factor Zap1, a central mediator of zinc homeostasis that also controls the expression of multiple adhesion genes. Similarly, Kalinina et al. [32] proposed that the enhanced adherence of Goliath cells may partly result from increased cell surface hydrophobicity, representing an adhesion pathway distinct from that mediated by hyphae, whose development is delayed in this form.

Notably, Goliath cells adhere more effectively to oral and vaginal epithelial cells, as well as abiotic substrates such as plastics, underscoring their potential relevance in biofilm formation on medical devices. Although their precise role in pathogenesis remains unresolved, their distinctive features point to a specialized contribution to biofilm persistence and host colonization.

The mechanisms mentioned above, together with the densely packed cell layers, contribute to the remarkable resistance abilities of *Candida* spp. biofilms. As a result, infections and colonization caused by these biofilm-forming fungi are particularly challenging to manage. In this study, we present a novel approach that may strengthen efforts to combat such biofilms.

## 4. Nanomaterials Against *Candida* spp. Biofilms

Nanomaterials are materials with at least one dimension in the nanometer range (1–100 nm), which gives them unique physical, chemical, and biological properties compared to their bulk counterparts [33]. Nanomaterials have recently emerged as highly promising antifungal platforms for the management of *Candida* spp. biofilms, which are notoriously resistant to conventional therapeutics. Their nanoscale dimensions confer unique properties, including enhanced surface reactivity, biofilm penetration, and controlled drug release, enabling both direct antifungal action and improved delivery of existing agents [34].

Several distinct classes of nanomaterials have been explored in this context. Nanoparticles, especially metallic and metal oxide nanoparticles, particularly silver, gold, zinc oxide, and titanium dioxide and transition metal oxides, are among the most extensively studied, demonstrating potent biofilm inhibition through reactive oxygen species (ROS) generation and membrane disruption. Carbon dots (CDs) can also penetrate Candida biofilms, deliver antifungal agents, and generate ROS while remaining biocompatible. Equally important are nanoscale delivery systems. Chitosan-based nanoparticles disrupt fungal membranes and inhibit adhesion. Liposomes have long been used to encapsulate antifungal agents, improving their penetration and reducing toxicity. Related lipid-based carriers, including solid lipid nanoparticles (SLNs) and nanocapsules, provide controlled release and stability advantages. Enzyme-mimicking nanoenzymes further contribute by catalytically degrading extracellular biofilm components and producing fungicidal radicals. Table 1 provides an overview of the key advantages and limitations of the aforementioned nanomaterials, as well as their potential applications in photocatalytic strategies against *Candida* biofilms.

Potential shapes of nanomaterials include nanoparticles (spherical), nanorods (cylindrical), nanowires, nanosheets (flat), nanoporous structures, nanotubes (tubular), Janus nanoparticles (asymmetric), core–shell nanoparticles, nanostars (star-shaped), nanocubes, and nanoflowers (flower-like). The shape of a nanomaterial influences its ability to combat *Candida* biofilms by enhancing adhesion, penetration, and direct interaction with fungal cells; for example, nanorods and nanoflowers, with their large surface area and sharp edges, can more effectively disrupt the biofilm structure, facilitate deeper drug or reactive species delivery, and achieve stronger antifungal effects compared to spherical nanoparticles [35,36,37].

**Table 1 molecules-30-04500-t001:** An overview of the principal advantages and inherent limitations of nanomaterials used for *Candida* biofilm inactivation, along with their prospective applications in photocatalytic interventions targeting these biofilms.

Nanomaterial Type	Advantages	Limitations	Applied in Photocatalysis Against Biofilm *Candida* spp. —Subsection	References
Metallic nanoparticles	Multiple mechanisms of action (ROS generation, membrane and DNA damage), reducing the risk of resistance. Can act as drug carriers, enhancing the efficacy. Effective as surface coatings on medical devices. Prevent the reformation of biofilms.	Potential cytotoxicity to human cells and environmental accumulation. Stability issues and unpredictable ion release. Limited long-term effectiveness.	5.2, 5.3, 5.6	[38]
Metal oxide nanoparticles	Use multiple mechanisms that disrupt cell structures and virulence factors, reducing resistance risk. Stable incorporation into medical materials. Synergistic action with antifungal drugs. Prevent initial fungal adhesion and long-term biofilm colonization.	Cytotoxicity and oxidative stress to potential human host cells. Tend to aggregate in physiological fluids and within biofilm matrices. Lack of in vivo validation.	5.1, 5.2, 5.3, 5.6	[39,40]
Carbon dots	Effective inhibition of adhesion and biofilm formation. Low cytotoxicity toward potential host cells. Surface tunable—CDs can be modified to boost antifungal activity and stability. Photostable and multicolor—CDs resist photobleaching and enable real-time biofilm detection.	Less effective against fully developed biofilms. Requires surface optimization.	5.5	[41,42,43]
Chitosan-based nanoparticles	Chitosan is a natural, biodegradable polymer with low toxicity, suitable for biomedical applications. Can be loaded with antifungal agents and functionalized for targeted delivery. The positive charge of chitosan nanoparticles facilitates adhesion to negatively charged fungal cell membranes	Stability can be affected by pH and enzymatic degradation. Multispecies biofilms can reduce chitosan nanoparticle efficacy.	Not yet, only possible as a photocatalyst carrier.	[44,45,46,47]
Liposomes	Lipophilic lipid shell facilitates interaction with fungal membranes and biofilm matrix, enhancing penetration. Can act as a drug carrier, ensuring high drug concentration and protection from degradation. Can bypass resistance mechanisms such as efflux pumps. Biocompatibility and reduced toxicity.	Not for use as photocatalysts alone. High cost and complex production hinder clinical scaling of stable liposomes.	Not yet, only possible as a photocatalyst carrier.	[48,49,50,51]
Solid lipid nanoparticles (SLNs)	Efficient penetration of the biofilm matrix. Remain stable across different pH and temperatures. Biocompatibility and reduced toxicity. Can function as antifungal drug delivery system, maintaining high levels.	Not for use as photocatalysts alone. May become unstable during storage due to aggregation, fusion, or drug loss. Can lose stability due to polymorphic transitions. Manufacturing entails high costs.	Not yet, only possible as a photocatalyst carrier.	[52]
Nanoenzymes	Ideal source of ROS. Can act like DNase, breaking down DNA and disrupting biofilms. Possibly combined with probiotics for enhanced antifungal effects. Easy to synthesize.	Catalytic activity can be unstable within biofilms due to environmental fluctuations.	5.4	[53,54,55,56]

Recent advances in nanotechnology have enabled the development of more sophisticated nanomaterials that also offer effective strategies against *Candida* biofilms.

Advanced nanostructures, such as nanorobots, are designed to actively penetrate dense *Candida* biofilm matrices and enhance the delivery of drugs or photocatalytic agents. Nanorobots can mechanically disrupt the biofilm structure while delivering antifungal drugs or photocatalysts directly to fungal cells. They can also generate ROS or trigger photodynamic effects to kill the cells. Some nanorobot designs are capable of sensing the biofilm environment and releasing drugs precisely where needed, improving treatment specificity and efficacy [57].

Similarly, nanomotors offer complementary biofilm penetration, acting as nanoscale devices capable of autonomous movement that facilitates active penetration and mechanical disruption of *Candida* biofilm architecture. This motility overcomes a major barrier in biofilm-associated infections, enabling therapeutic agents to reach deeper layers that are typically inaccessible to conventional treatments. Beyond mechanical action, nanomotors can be functionalized for multimodal therapy, carrying chemical antifungal agents or incorporating photodynamic/photocatalytic components that generate ROS to eradicate fungal cells and biofilm. Some designs also permit spatiotemporally controlled drug release, responding to local biofilm cues to maximize therapeutic specificity while minimizing off-target effects [58].

Nanoemulsions demonstrate high efficacy against *Candida* spp. biofilms due to their unique nanoscale droplet structure, which allows them to penetrate the dense ECM that often impedes conventional therapies. Their small droplet size (typically below 200 nm) and appropriately formulated oil phase and surfactants enable efficient passage through the biofilm’s polysaccharide barrier, facilitating direct delivery of antimicrobial agents to fungal cells. Additionally, their structure allows for controlled release of active compounds, enhancing antifungal effectiveness [59].

Together, these diverse nanomaterials with more advanced mechanisms of action represent a rapidly expanding arsenal against *Candida* spp. biofilms, combining intrinsic antifungal effects with enhanced delivery and biofilm-disrupting capabilities, highlighting their potential as antifungal therapeutics.

Early-stage inhibition of biofilm formation is equally crucial in managing *Candida* infections, as it can significantly reduce infection risk. Strategies such as nanocoatings have been developed, showing promise in combating *Candida* biofilms and offering an effective alternative to conventional antifungal therapies. Their large surface area and chemical modifiability allow interactions with medical and biological surfaces, inhibiting cell adhesion and biofilm formation. They can penetrate existing biofilms to deliver antimicrobial agents directly to fungal cells. Some nanocoatings, such as TiO_2_, exhibit photocatalytic activity, generating ROS under light to kill *Candida* cells. Applied to medical devices like catheters and implants, nanocoatings can reduce infection risk and biofilm development, highlighting their potential in antifungal strategies [60,61].

In addition to their intrinsic antifungal properties, numerous nanomaterials exhibit exceptional potential for photocatalytic applications, leveraging their nanoscale characteristics to generate ROS upon irradiation [62]. For example, metallic and metal oxide nanoparticles—including TiO_2_, ZnO, and Ag-based nanostructures—as well as carbon-based nanomaterials such as CDs, display tunable electronic and optical properties that promote efficient charge separation and transfer under UV or visible light [63].

In future perspectives, special attention should be given to the development of intelligent nanomaterials capable of responding to specific biological stimuli, such as pH or enzymatic activity, to enable targeted and on-demand antimicrobial action. Such stimuli-responsive systems can remain inactive under physiological conditions and become activated only in the microenvironment of *Candida* biofilms, where pH decreases or enzymatic activity increases, thus enhancing selectivity and minimizing cytotoxicity [64]. In parallel, recent advances in nanotechnology for fungal biofilms underscore the promise of nanocarriers and multifunctional nanoparticles to penetrate dense fungal matrices, deliver antifungal agents, generate ROS, and respond to microenvironmental cues. Such dual-function nanoplatforms can be engineered to remain inert under normal conditions but rapidly activate—via pH drop, enzyme over-expression, or ROS triggers—when encountering *Candida* biofilms. By integrating stimulus-responsiveness with photocatalytic or photodynamic elements, these systems hold the potential to achieve localized fungal inactivation with minimal host damage. Ultimately, targeting the unique microenvironment of *Candida* biofilms opens a path toward smarter, adaptive antifungal therapies embedded in coatings, implants, or wound-care materials [49,65,66].

Another promising approach is hybrid therapy combining photocatalysis with photodynamic therapy (photocatalytic–photodynamic combined therapy, P–PCT), which offers an effective strategy for combating *Candida albicans* biofilms. Photodynamic therapy (PDT) utilizes a photosensitizer that, upon activation by light of a specific wavelength, produces ROS, such as singlet oxygen and hydroxyl radicals (•OH), which damage fungal cells through oxidation of proteins, lipids, and nucleic acids, ultimately leading to apoptosis or cell lysis. In hybrid P–PCT systems, such as Ag-doped TiO_2_ nanoparticles, synergistic effects occur: photocatalytic ROS generation under visible light helps degrade the biofilm matrix, enhancing the penetration and activity of the photosensitizer and improving PDT efficacy [67]. Another example is oligo-chitosan-modified nanoparticles loaded with protoporphyrin IX (OC-PpIX NP), which produce ROS upon light activation, effectively penetrate dense biofilm structures, and kill fungal cells [68]. These hybrid systems allow selective “on-demand” activation, minimizing cytotoxicity toward host tissues. Such strategies open the way for designing intelligent medical coatings, implantable devices, or wound-care materials capable of effectively eradicating *Candida* spp. biofilms, even in hard-to-reach areas where conventional antifungal treatments are often ineffective.

The resulting photogenerated ROS can inflict oxidative damage on fungal cell walls, plasma membranes, and the EPS, thereby facilitating biofilm disruption and microbial inactivation. Moreover, the elevated surface-to-volume ratio inherent to these nanomaterials also enhances photon absorption and interaction with biofilm constituents, enabling spatially confined and controllable photocatalytic activity. By coupling intrinsic antimicrobial effects with light-activated ROS generation, nanomaterials offer a dual-mode strategy for the eradication of *Candida* spp. biofilms, addressing their pronounced resistance to conventional antifungal therapeutics [69,70]. The utilization of nanomaterials capable of mediating photocatalytic elimination of *Candida* biofilms is comprehensively discussed in Section 6.

## 5. Photocatalysis Against *Candida* spp. Biofilms

Photocatalytic activation, as mentioned before, induces the generation of ROSs, which elicit profound oxidative stress in *Candida* spp., targeting membrane lipids, mitochondrial proteins, and nucleic acids, thereby compromising structural integrity and cellular homeostasis. The ensuing lipid peroxidation, protein oxidation, and nucleic acid damage culminate in loss of membrane functionality, respiratory chain collapse, and irreversible impairment of replication and transcriptional processes. Simultaneously, photocatalytic reactions degrade critical EPS constituents and oxidatively inactivate adhesins such as ALS family proteins, disrupting intercellular cohesion and surface attachment. Through the combined destabilization of both cellular and extracellular targets, photocatalysis effectively dismantles the protective architecture of the biofilm, enabling its inactivation.

### 5.1. Photocatalysis

Photocatalysis harnesses photon energy to initiate and regulate chemical transformations on non-adsorptive substrates through fundamental mechanisms such as single-electron transfer, energy transfer, or atom transfer. The efficiency of these processes critically depends on the nature of the photocatalyst (PC), which may be a metal complex, an organic chromophore, or another functional material capable of absorbing light and mediating charge or energy flow. The term “photocatalyst” reflects this dual function: “photo” denotes activation by light, while “catalyst” refers to a substance that alters reaction kinetics without being consumed. Upon irradiation, photocatalysts promote the formation of reactive excited states that lower energetic barriers, enabling otherwise unfavorable reactions to proceed under mild and controlled conditions [71].

The photocatalytic activity of photocatalyst is intrinsically linked to their capacity to generate electron–hole (e^−^/h^+^) pairs, which act as the primary agents driving radical formation. These reactive species subsequently initiate a sequence of redox reactions that constitute the core of the photocatalytic mechanism [72]. When the photocatalyst is irradiated with photons possessing energy equal to or greater than its band gap—the energy difference between the valence band (VB) and the conduction band (CB)—electrons in the VB are excited to the CB, leaving behind positively charged holes (h^+^) in the VB. This photoexcitation event triggers the catalytic process, contingent upon sufficient photon absorption to sustain continuous generation of charge carriers. Once formed, the photogenerated electrons and holes migrate to the surface of the photocatalyst, where they participate in redox reactions. Holes (h^+^), characterized by strong oxidative potential, interact with adsorbed water molecules or hydroxide ions to produce •OH, among the most potent oxidants in photocatalysis. These radicals propagate chain reactions capable of degrading diverse organic substrates. Simultaneously, the excited electrons (e^−^), possessing strong reducing properties, are transferred to electron acceptors such as molecular oxygen, generating ROS like superoxide anions (O_2_•^−^), or may reduce metal ions on the catalyst surface, potentially depositing in a lower oxidation state. Overall, the efficiency of photocatalysis is determined by the interplay of light absorption, charge carrier generation, and surface redox chemistry [73,74] (Figure 2).

Photocatalysts can be divided into several main groups. Semiconductor photocatalysts include metal oxides (TiO_2_, ZnO, WO_3_), sulfides (CdS, MoS_2_), and nitrides and carbides (BN, SiC), valued for their stability and ability to generate electron–hole pairs. Transition metal complexes, such as polypyridyl complexes of Ru, Ir, Cu, and Co and Mn complexes, show high efficiency in photocatalytic reactions, particularly under visible light. Organic photocatalysts include porphyrins, phthalocyanines, conjugated polymers, and dyes like eosin Y, used, for example, in photodynamic therapy. Hybrid and composite photocatalysts combine semiconductors with metals (TiO_2_–Au) or carbon-based materials (TiO_2_–graphene), improving charge separation and material stability. Additionally, biohybrid systems, such as liposomes, polymeric nanoparticles, and dendrimers, facilitate the delivery of photocatalysts [75,76,77,78].

### 5.2. Mechanism of Photocatalysis Against Candida spp. Biofilm

Photocatalysis under light irradiation activates a semiconductor, resulting in the generation of ROS, including •OH, O_2_•^−^ and hydrogen peroxide (H_2_O_2_). These ROS exhibit high reactivity and can infiltrate the cell membrane of *Candida* spp., where they target the unsaturated fatty acid moieties within membrane lipids, initiating lipid peroxidation [79]. This process leads to the formation of lipid hydroperoxides, which compromise the structural integrity of the lipid bilayer, generating membrane pores and microlesions that permit the efflux of ions and small metabolites [80]. Concurrently, ROSs induce oxidative modifications of membrane-associated proteins, including ion channels and receptors, thereby disrupting transmembrane transport and overall membrane architecture. The cumulative degradation of lipids and proteins diminishes membrane fluidity and mechanical stability, resulting in depolarization and collapse of the cellular ion gradient, ultimately impairing cellular homeostasis and viability [81].

ROSs penetrate the cell membrane and biofilm matrix, reaching the nucleus and cytoplasm, where they directly interact with nucleic acids, including DNA and RNA, •OH attack the deoxyribose backbone of DNA, causing single- and double-strand breaks, and also induce oxidative modifications of nitrogenous bases, particularly guanine (formation of 8-oxoguanine), leading to mutagenesis and replication errors. ROS can also oxidize phosphodiester bonds in the DNA and RNA backbone, destabilizing the helical structure and disrupting hydrogen bonding between bases. In RNA oxidative modifications of nucleotides occur, resulting in destabilization of secondary structures and loss of functionality in translation processes. Free radicals can additionally react with lipid peroxidation products in the membrane, generating reactive aldehydes, e.g., MDA, that further modify DNA and RNA through base adduct formation. Accumulated DNA and RNA damage activates intracellular repair mechanisms, such as single- and double-strand break repair pathways, but excessive ROS overwhelms these systems, leading to their failure. Disruption of nucleic acid integrity halts replication, transcription, and translation, resulting in impaired synthesis of proteins and enzymes critical for metabolism. Ultimately, the accumulation of oxidative DNA and RNA damage triggers apoptotic and necrotic cell death pathways, contributing to cell inactivation and biofilm eradication of *Candida* spp. [82,83,84].

Furthermore, ROS penetrate the cytoplasm and react with cellular proteins, leading to oxidative modifications that primarily affect membrane-associated enzymes and structural proteins. Oxidation induces conformational changes in proteins, causing denaturation and loss of enzymatic activity, which disrupts essential metabolic processes and impairs the transport of metabolites across the cell membrane. Structural proteins in the cell wall and biofilm matrix, such as glucan- and chitin-associated proteins, are particularly vulnerable, resulting in destabilization of the biofilm architecture and weakening of its protective barrier. Mitochondrial proteins involved in oxidative phosphorylation, including components of the electron transport chain, are also targeted, leading to reduced ATP production and disruption of cellular energy homeostasis. ROS can promote protein aggregation and formation of cross-links, further impairing protein function. The accumulation of oxidized and misfolded proteins triggers stress responses, but cellular repair systems, such as molecular chaperones and proteasomes, are often overwhelmed. This imbalance contributes to progressive cellular dysfunction, loss of membrane integrity, and failure of metabolic pathways [83,85].

Photocatalysis mediates the degradation of the polysaccharide components of the EPS, particularly β-glucans and mannans, which, as mentioned before, are essential for the cohesion and viscoelastic properties of the biofilm. Through photocatalytic reactions, matrix-associated proteins, including enzymes and adhesins, undergo oxidative modifications that compromise the capacity of *Candida* cells to interconnect and adhere to surfaces. The photocatalytic process also disrupts lipid and glycoprotein constituents, destabilizing the EPS network and enhancing its permeability. In addition, photocatalysis targets extracellular DNA (eDNA), a crucial structural element of the EPS, leading to nucleic acid oxidation, loss of integrity, and disruption of biofilm cohesion. eDNA in *Candida* spp. biofilms stabilizes structure, enhances adhesion, and increases drug resistance. As the EPS undergoes progressive photocatalytic degradation, the biofilm loses its protective barrier, thereby facilitating the penetration of antifungal agents and host immune defenses [86,87,88].

ROS generated during photocatalysis oxidize cell-surface adhesins, such as ALS agglutinins, leading to protein denaturation and loss of receptor-binding capacity. Other adhesins and invasins are similarly impaired, thereby disrupting both initial colonization and biofilm maturation. Photocatalysis also alters cell-surface hydrophobicity and charge, reducing the efficiency of intercellular interactions. In addition, it downregulates the expression and functional activity of adhesins, further limiting the ability of *Candida* cells to maintain stable attachment. As noted previously, photocatalytic degradation of EPS components amplifies this effect by weakening cell–cell interactions. Collectively, these alterations result in impaired adhesion, cell dispersal, and destabilization of biofilm architecture [89].

In summary, photocatalytic activation generates ROS that induce oxidative stress in *Candida* spp., damaging membrane lipids, mitochondrial proteins, and nucleic acids. This leads to membrane dysfunction, respiratory chain collapse, and impaired genetic processes. At the same time, photocatalysis degrades EPS and inactivates adhesins such as ALS proteins, weakening cell–cell cohesion and surface attachment. Together, these effects disrupt biofilm structure and promote its inactivation. All the major effects of photocatalysis on biofilm *Candida* spp. have been summarized in Figure 3.

### 5.3. Candida spp. Response Mechanisms to Oxidative Stress Induced by ROS

Although research in this area is still evolving, studies have investigated the mechanisms by which *Candida* spp. respond to oxidative stress caused by reactive oxygen species (ROS) generated during photocatalytic eradication processes.

A study by Arribas et al. [90] highlighted novel insights into the oxidative stress response of *Candida albicans*. Three primary signaling mechanisms mediate this response: MAPK cascades, transcriptional regulation, and DNA damage checkpoints, which together coordinate antioxidant defenses and repair processes. Upon encountering oxidative stress, *C. albicans* senses ROS and rapidly initiates a multifaceted defense program through these signaling pathways. The yeast employs multiple antioxidant systems to neutralize ROS: catalase converts hydrogen peroxide into water and oxygen, while the thioredoxin and glutathione systems reduce oxidized proteins and detoxify reactive molecules, with NADPH produced via the pentose phosphate pathway providing the necessary reducing power. Simultaneously, *C. albicans* reorganizes its proteome, increasing the abundance of antioxidant enzymes, heat shock proteins, and other metabolic proteins to enhance survival under oxidative conditions. ROS-induced DNA damage activates repair mechanisms, including base excision repair, nucleotide excision repair, and homologous recombination, thereby preserving genome integrity. Misfolded or damaged proteins are degraded through proteasome activation to prevent toxic accumulation, while enzymes involved in lipid metabolism and membrane remodeling maintain membrane integrity and fluidity. These responses are tightly regulated by transcription factors such as Cap1 and Hap43, allowing the yeast to dynamically adjust its defenses. Collectively, these coordinated strategies enable *C. albicans* to withstand oxidative stress, repair cellular damage, and remain viable during interactions with host immune cells.

In *C. albicans*, Arribas et al. [91] also highlighted the potential role of the Prn1 protein in oxidative stress response. Although its exact function is unknown, Prn1 shares similarities with mammalian Pirin, whose expression is linked to activation of the antioxidant transcription factor Nrf2 and downstream proteins such as NAD(P)H oxidoreductase 1 (NQO1). Under ROS-induced oxidative stress, Prn1 levels increase, supporting detoxification and enhancing cell survival and recovery. In Prn1-deficient cells, proteasome activity rises, translation-related proteins decrease, and several transcription factors show altered abundance, indicating Prn1’s central role in coordinating the oxidative stress response.

A study by Briones-Martin-Del-Campo et al. [92] explored a similar topic, focusing on the oxidative stress response in *Candida glabrata*. This yeast employs multiple strategies to cope with oxidative stress induced by ROS. *C. glabrata* detoxifies ROS through enzymatic defenses, including catalase (CgCta1) and superoxide dismutases (CgSOD1 and CgSOD2), as well as non-enzymatic systems such as glutathione and thioredoxin. Additional protective mechanisms involve pigment production and metal-binding proteins. The expression of these defense systems is regulated by transcription factors, including Yap1, Skn7, Msn2, and Msn4, allowing *C. glabrata* to maintain redox homeostasis and survive under oxidative stress conditions.

These mechanisms illustrate how members of the *Candida* spp. family cope with oxidative stress induced by ROS, for instance, during photocatalytic reactions. Although the full understanding of these oxidative stress response systems is still developing, and further research is needed for some *Candida* species, the described factors demonstrate how *Candida* spp. can potentially mitigate ROS-induced damage and reduce the effectiveness of oxidative eradication strategies. This area also represents an interesting opportunity for the development of novel antifungal agents aimed at inhibiting the oxidative stress response, thereby enhancing the effects of ROS on the fungi.

## 6. Methods of Photocatalytic Inactivation and Eradication of *Candida* spp. Biofilms

This section presents the current state of knowledge regarding the application of nanomaterials as photocatalysts in the eradication of *Candida* biofilms. Photocatalytic nanomaterials generate ROS upon light exposure, and as detailed in the section on the effects of photocatalysis on *Candida*, ROS damage fungal cells and destabilize the biofilm matrix, enabling not only the elimination of planktonic *Candida* cells but also the disruption of its mature biofilms. This review further addresses strategies for structural and doping modifications of nanomaterials that enhance their antifungal activity. For instance, surface functionalization of nanoparticles or the incorporation of specific metal ions can improve material stability and increase ROS production. In recent years, considerable attention has been given to nanozymes—nanoparticles exhibiting enzymatic activity—that can degrade components of the biofilm matrix and thereby potentiate the effects of antifungal drugs. Equally significant is the development of hybrid and surface-modified materials, which combine multiple mechanisms of action, including their photocatalytic properties, to enhance anti-biofilm efficacy. Such approaches are particularly relevant for clinical applications, including the fabrication of antibacterial coatings for implants, catheters, and other medical devices, as well as the development of novel therapies targeting recalcitrant fungal infections. While the application of nanotechnology in combating *Candida* biofilms in medical settings shows great promise, most of these strategies still require extensive large-scale studies. Nonetheless, they represent highly promising avenues, especially in the context of rising healthcare-associated infections and the increasing prevalence of multidrug-resistant hospital-acquired microbial flora.

### 6.1. Titanium Dioxide: The Foundation of Nanomaterials Research Against Candida Biofilms

Titanium dioxide (TiO_2_) is among the most well-established and widely utilized photocatalysts. Owing to its strong photoreactivity, antimicrobial properties, and self-cleaning ability, TiO_2_ has attracted considerable attention as a material for preventing biofilm formation. TiO_2_ exists in several polymorphic crystalline forms, with the three most common being anatase, rutile, and brookite. Anatase, the most photoactive form of TiO_2_, is stable at room temperature but can convert to rutile at high temperatures. Rutile is thermodynamically stable with lower photocatalytic activity and is widely used as a white pigment. Brookite is rare, more difficult to synthesize, but also exhibits photocatalytic activity, typically in nanostructured form. At present, the majority of advanced research on this photocatalyst is devoted to modified and doped forms of TiO_2_, which enable a red-shift in the absorption edge and thus broaden its activity into the visible-light region. The use of TiO_2_ nanoparticle-based photocatalysis is a well-established strategy for addressing *Candida* spp. biofilms in various fields [93].

A study by Helmy et al. [94], which introduced a novel TiO_2_ synthesis method using *Malva parviflora*, also evaluated the photocatalytic antimicrobial properties of TiO_2_ nanoparticles against *Candida albicans*. In this work, the nanoparticles were S-doped TiO_2_ NPs, synthesized via the sol–gel process with an aqueous extract of *Malva parviflora*, serving as a green reducing and stabilizing agent to promote sulfur doping and anatase crystal formation. The antimicrobial activity of these nanoparticles was tested against various microorganisms, including *C. albicans*, under an LED board (30 W, λ = 450 nm) for 8 h, with a 10 cm distance between the sample and the light source. Upon irradiation, S-doped TiO_2_ NPs produce ROS such as •OH and H_2_O_2_, which can penetrate fungal cells and trigger oxidative stress, achieving over 90% inhibition of *C. albicans*. Although the study did not evaluate the effect on pre-formed *C. albicans* biofilms, the results demonstrate significant potential of S-doped TiO_2_ against yeast cells and suggest promising applications in preventing biofilm formation.

Tzeng et al. [95] investigated the photocatalytic properties of nitrogen-doped TiO_2_ (N-TiO_2_) and tourmaline–nitrogen co-doped TiO_2_ (T-N-TiO_2_) against *C. albicans* under visible light irradiation. N-doped TiO_2_ was obtained via a modified sol–gel process, where titanium isopropoxide, ammonium hydroxide, ethanol, and deionized water were reacted, followed by calcination at 500 °C for 2 h. The material crystallized entirely in the anatase phase. To prepare the T-N-TiO_2_ composite, tourmaline powder (4 wt%) was mixed with the N-TiO_2_ and the mixture was again heated at 500 °C for 2 h, also yielding pure anatase. Following their preparation, the photocatalysts were evaluated for antimicrobial activity against four different pathogens under visible light exposure. Among the tested microorganisms (*Staphylococcus aureus*, *Escherichia coli*, *Mycobacterium avium* and *Candida albicans*), *C. albicans* exhibited the highest resistance. Nevertheless, both the modified N-TiO_2_ and T-N-TiO_2_ demonstrated significant photocatalytic activity, effectively producing antifungal effects and highlighting their potential as visible-light-driven antimicrobial agents. Furthermore, enhancing visible-light intensity and raising the photocatalyst dose markedly boosted the inactivation performance of the system.

Another study by Nguyen et al. [89] proposed a similar photocatalytic strategy using TiO_2_-coated materials against *C. glabrata*. Polydimethylsiloxane (PDMS) plates were prepared with varying TiO_2_ concentrations, and photoactivation was induced by LED exposure. The LED lamp operated at 8.3 W, 120 VAC, 60 Hz, and 0.07 A. The experiment was conducted for 24 h under both dark and light conditions. The results demonstrated that the strongest inhibition of *C. glabrata* biofilms occurred on PDMS plates containing 5% TiO_2_ by weight under irradiation. While the most pronounced effects were achieved when the coating was combined with plasma treatment, and most antibiofilm activity was associated with the surface properties of the material (particularly hydrophobicity), the findings clearly indicated that TiO_2_-based photocatalysis alone significantly inhibited *C. glabrata* biofilms—a species known for its high resistance to oxidative stress. Nevertheless, the combined approach yielded the greatest overall effect.

In recent years, an increasing number of studies have investigated various combinations involving TiO_2_ nanoparticles (NPs) for photocatalytic applications against *Candida* spp. Although further research is needed to specifically evaluate their effectiveness against mature *Candida* biofilms, this photocatalytic approach shows significant potential.

### 6.2. Transition Metal Oxide Nanoparticles: Structural Modifications and Doping Approaches

Transition metal oxides (TMOs) are semiconducting materials capable of absorbing photons to initiate redox reactions, rendering them highly efficient photocatalysts. Representative TMOs include transition metal oxides such as iron oxide (Fe_2_O_3_), copper oxide (CuO), cobalt oxide (CoO), nickel oxide (NiO), and manganese oxides (MnO, MnO_2_), which exhibit characteristic electronic and catalytic properties. Their photocatalytic activity is primarily attributed to the generation of electron–hole pairs upon light irradiation, which drive oxidative and reductive processes that decompose complex organic molecules. When engineered at the nanoscale, TMOs exhibit significantly enhanced performance due to shortened charge carrier diffusion lengths, facilitating more efficient electron–hole migration and improving overall photoactivity. Furthermore, these nanostructured materials combine chemical stability, high surface reactivity, and environmental compatibility, making them versatile for diverse photocatalytic applications [96]. Main strategies to enhance the effectiveness of combating *Candida* spp. biofilms using transition metal oxides as photocatalysts are summarized in Figure 4.

Nanostructured TMOs have also demonstrated considerable potential for combating *Candida* spp. biofilms. Upon light activation, these nanomaterials produce ROS capable of penetrating the extracellular polysaccharide–protein matrix, inducing oxidative damage to fungal cell membranes, proteins, and nucleic acids. The high surface area and nanoscale morphology of these materials promote intimate interactions with fungal cells, disrupting adhesion mechanisms and compromising biofilm structural integrity. Consequently, TMO-based nanostructures offer a robust photocatalytic strategy for the inactivation of both planktonic *Candida* cells and their recalcitrant biofilm communities [97].

Engineering transition metal oxides into nanostructured morphologies, such as nanorods, nanowires, or nanosheets, significantly enhances their photocatalytic and antimicrobial performance. The high aspect ratio of nanorods facilitates directional charge transport and efficient separation of photogenerated electron–hole pairs, minimizing recombination and thereby improving photocatalytic efficiency. Furthermore, the increased surface-to-volume ratio of these nanostructures provides a greater density of active sites, enhancing interactions with microbial cells and biofilms and amplifying antimicrobial and antibiofilm effects. Consequently, the rational design of TMOs into specific nanostructures represents a powerful strategy to maximize their functional performance in photocatalysis and microbial control [98,99].

Jeba et al. [100] proposed enhancing the activity of ZrO_2_ by optimizing its morphology and conducted a study in which tetragonal zirconia (t-ZrO_2_) nanorods were synthesized via a co-precipitation method. The nanorods exhibited a rod-shaped morphology, high crystallinity with an average crystallite size of 29.74 nm, a wide band gap of 4.6 eV, and abundant surface oxygen vacancies, which collectively improved their photocatalytic and antimicrobial performance. Under UV light irradiation, the t-ZrO_2_ nanorods achieved 80% degradation of methylene blue within 180 min, demonstrating strong photocatalytic activity. Antifungal assays against *C. albicans* and *C. parapsilosis* showed inhibition zones of 25 mm and 18 mm, respectively, surpassing the efficacy of the standard antifungal Nystatin. The photocatalytically generated ROS, together with the high surface area of the nanorods, disrupted fungal cell membranes and biofilm structures, effectively reducing planktonic cell viability and compromising biofilm integrity. These results highlight t-ZrO_2_ nanorods as promising multifunctional materials for photocatalytic applications and for the control of *Candida* infections, including the potential eradication of biofilms.

Although there are currently no reports in the literature on the use of such nanomaterials for photocatalytic applications against *Candida* spp., this approach represents a promising research direction. Previous studies have demonstrated the effectiveness of transition metal oxides in nanostructured morphologies, such as nanorods, nanowires, nanosheets, or nanoflakes, against *Candida* and its biofilm [101,102,103]. Therefore, employing these already established nanomaterials as photocatalysts could offer a dual mode of action—combining the intrinsic antimicrobial properties of the nanomaterial with photocatalytic activity—potentially enhancing the overall efficacy of *Candida* eradication.

Doping in transition metal oxide nanoparticles has proven to be an effective strategy for tuning their structural, optical, and photocatalytic properties. The incorporation of suitable dopants enhances visible-light absorption, suppresses charge-carrier recombination, and promotes the generation of ROS. Such modified nanomaterials have the potential to exhibit significant efficacy against both planktonic fungal cells and biofilms, disrupting their structural integrity and defense mechanisms. Therefore, controlled doping represents a promising approach for the development of advanced nanomaterials with enhanced performance in combating *Candida* biofilms.

Nasr et al. [104] as part of their study synthesized zinc-doped manganese(III) oxide (Zn–Mn_2_O_3_) nanoparticles via a controlled precipitation method with Zn contents of 3%, 5%, and 10%, and specifically evaluated their antifungal properties against *C. albicans*. Structural and optical analyses revealed that Zn incorporation induced a phase transition from cubic Mn_2_O_3_ to tetragonal ZnMn_2_O_4_, accompanied by a reduction in the band gap from 2.26 eV in pure Mn_2_O_3_ to 1.89 eV in the 10% Zn-doped sample. These modifications enhanced visible-light absorption and altered nanoparticle morphology, thereby improving photocatalytic performance. In this phase of the research, antifungal assays under visible-light irradiation demonstrated a clear dependence of activity on Zn concentration. The 5% and 10% Zn–Mn_2_O_3_ samples exhibited the highest inhibitory effects against *C. albicans*, with inhibition zones of 4.0 ± 0.00 mm and 4.5 ± 0.10 mm, respectively. The 3% Zn sample showed moderate activity (3.5 ± 0.10 mm), whereas undoped Mn_2_O_3_ displayed negligible antifungal effects. These findings confirmed that Zn doping plays a decisive role in enhancing the antimicrobial efficiency of Mn_2_O_3_. The antifungal mechanism was attributed to the synergistic action of Zn^2+^ ion release and photocatalytically generated ROS. Zn^2+^ ions destabilized fungal membranes and interfered with essential enzymatic pathways, while ROS induced oxidative damage to lipids, proteins, and nucleic acids. Together, these processes effectively suppressed fungal growth and viability. Crucially, this segment of the study also demonstrated that Zn–Mn_2_O_3_ nanoparticles can act against *C. albicans* biofilms. ROS penetrated the extracellular polysaccharide–protein matrix, weakening its cohesion and exposing embedded cells to oxidative stress, while Zn^2+^ ions impaired adhesion and disrupted intercellular communication. This dual mechanism underscores the potential of Zn–Mn_2_O_3_ nanoparticles as candidates for targeting both planktonic and biofilm-associated forms of *Candida.*

Pugazhendhi et al. [105] investigated the photocatalytic properties and antimicrobial efficacy of Fe-doped CuO nanoparticles. The nanoparticles were synthesized via a sol–gel method, resulting in rectangular, agglomerated particles with an average crystallite size of 21 nm. Structural analysis confirmed the monoclinic CuO phase, and EDAX analysis verified homogeneous Fe incorporation. Optical characterization revealed a narrowed band gap of 1.0 eV, which enhanced visible-light absorption and photocatalytic performance. The antifungal and antibiofilm activities were evaluated, including against *C. albicans* under visible-light irradiation. Generated ROS induced oxidative damage to fungal cell membranes, proteins, and nucleic acids, markedly reducing planktonic cell viability. Moreover, ROS penetrated the extracellular polysaccharide–protein matrix of the biofilm, while Fe ions disrupted adhesion and intercellular signaling, leading to biofilm destabilization. The photocatalytic effect was concentration-dependent, with the highest tested concentration (100 µg/mL) achieving 76.4% inhibition of *C. albicans* biofilm, demonstrating the potential of Fe–CuO nanoparticles as multifunctional agents for controlling fungal biofilms.

### 6.3. Zinc Oxide Nanostructures and Hybrid Systems in Combating Candida Biofilms

An alternative strategy to combat *Candida* spp. biofilms involves the application of zinc oxide (ZnO) nanostructures. ZnO, a semiconductor material, generates electron–hole pairs upon light exposure, which subsequently drive surface redox reactions. In typical photocatalytic processes, these reactions yield ROS capable of degrading pollutants and disrupting microbial structures. This highlights ZnO’s potential as an effective photocatalyst for both environmental and biomedical applications, including the eradication of *Candida* spp. biofilms [106].

Hameed et al. [107] investigated the photocatalytic properties of ZnO nanoparticles doped with metal ions (Mg^2+^, Ca^2+^, Sr^2+^, and Br^2+^) against *C. albicans*. Following synthesis, the doped nanoparticles (1 mg) were inoculated into plates containing *C. albicans* strains and incubated for 24 h at 30 °C under visible light. The most pronounced inhibitory effect was observed in the Mg-doped ZnO samples. At a minimum concentration of 2000 μg/mL, the Mg-doped ZnO nanoparticles completely inhibited *C. albicans* growth through ROS generation, demonstrating strong antifungal activity and the potential of this approach against *Candida* biofilms.

Saqib Saif et al. [108] examined ZnO in the form of flower-like nanostructures (“Nano-Flowers”) as a potential antimicrobial agent. ZnO was synthesized via two methods: a chemical route, yielding nanoparticles with an average size of 30 nm, and a plant-mediated (green phyto-synthesis) method, producing particles of approximately 25 nm. The photocatalytic activity of both types of ZnO Nano-Flowers was evaluated under sunlight through methylene blue degradation. The chemically synthesized ZnO NFs achieved up to 78% degradation, whereas the phyto-synthesized ZnO NFs reached 90%, confirming strong photocatalytic performance in both cases. Moreover, their antimicrobial activity was tested against *C. albicans*. The green-synthesized ZnO NFs exhibited complete (100%) inhibition, while the chemically synthesized counterparts achieved 80% inhibition. These findings, demonstrating strong antifungal activity, suggest that ZnO-based photocatalysis may represent a promising approach for the eradication of *Candida* spp. biofilms.

Arzate-Quintana et al. [109] performed SEM analysis to evaluate the effects of photocatalysis with a ZnO layer combined with UV light on the biofilm structure of *C. albicans*. Two groups of *C. albicans* biofilms with ZnO were observed: one was exposed to UV light for 60 min, while the other (control group) was not. The results demonstrated a pronounced biofilm inhibition in the UV-exposed group, where the biofilm was no longer detectable after 60 min, whereas the control group retained biofilm integrity.

Kim et al. [110] proposed a combination of ZnO nanoparticles (NPs) with biopolymer-based acrylic resins, specifically poly(methyl methacrylate) (PMMA). Although this material has been widely used in dental medicine, it is susceptible to microbial adhesion and subsequent biofilm formation. To address this, Kim et al. aimed to incorporate the photocatalytic properties of ZnO NPs to achieve antimicrobial effects in the modified acrylic resin. For this purpose, a 2 wt% silane-treated zinc oxide nanoflake (S-ZnNF)-incorporated PMMA was prepared. This composite was tested against *C. albicans* under UV light exposure for 30 min. The resulting ROS generation exhibited significant antimicrobial effects, demonstrating yet another sign of the potential of ZnO NPs as an additional strategy to combat biofilm-forming microorganisms such as *C. albicans*.

### 6.4. Nanozymes as a Novel Antifungal Concept

Nanozymes are nanomaterials with enzyme-like properties that can mimic the activity of natural enzymes by catalyzing biochemical reactions. They can be classified according to the type of catalytic activity they exhibit. Peroxidase-like nanozymes (POD-like) catalyze the decomposition of H_2_O_2_ to generate ROS, such as •OH, and are widely applied in antibacterial and antifungal therapies. Oxidase-like nanozymes (OXD-like) utilize molecular oxygen to oxidize substrates, which is useful in biosensing and anticancer applications. Catalase-like nanozymes (CAT-like) decompose H_2_O_2_ into water and oxygen, helping to reduce oxidative stress within cells, while superoxide-dismutase-like (SOD-like) nanozymes convert O_2_•^−^ into H_2_O_2_ and O_2_, protecting cells from oxidative damage. Less common nanozymes, such as hyaluronidase- or lipase-like types, catalyze specific biochemical reactions, including the degradation of macromolecules. Nanozymes differ in chemical composition and structure, and can be based on noble metals (e.g., Au, Cu), metal oxides (e.g., Fe_3_O_4_, MnO_2_), CDs, or hybrid nanostructures. Their stability, low cost, and tunable activity through chemical modifications or light exposure make them promising tools in medicine, including antifungal therapy and the eradication of biofilms [111]. By generating ROS, nanozymes disrupt the oxidative–reductive balance within fungal cells, damaging critical cellular components such as membranes, proteins, and nucleic acids. This disturbance in the redox system induces oxidative stress and, as mentioned before, leads to progressive fungal cell death and the inhibition of growth, making nanozymes an effective strategy for combating fungal pathogens, particularly in biofilm-associated infections [112].

The study conducted by Li et al. [113] reports the synthesis of an innovative nanozyme based on copper- and iodine-doped carbon dots (Cu/I-CDs) with strong antifungal and antibiofilm activity against *C. albicans*. The nanozyme exhibits peroxidase-like activity, enabling the catalytic decomposition of H_2_O_2_ and the generation of ROS, particularly •OH and O_2_•^−^, which effectively damage fungal cells and disrupt drug-resistant biofilms. Under conditions of 0.5 mM exogenous H_2_O_2_, a Cu/I-CD concentration of 0.585 mg/mL, and 90 min of photocatalysis using a 16 W LED, the survival rate of *C. albicans* was reduced to 10%, alongside significant antibiofilm effects. The antifungal efficacy of the nanozyme was confirmed both in vitro and in vivo, including in mouse wound and vaginal infection models, where accelerated wound healing and effective treatment of vulvovaginal candidiasis were observed. Importantly, Cu/I-CDs demonstrated high biocompatibility, with no significant toxicity in hemolysis and MTT assays and no adverse effects even after 60 days of application. These findings suggest that the synthesized Cu/I-CD nanozyme is a promising alternative to conventional antifungal drugs with broad potential for treating *C. albicans* infections.

A previous study focused on developing an antifungal system based on nitrogen- and iodine-doped carbon dots (I-CDs) that function as peroxidase-mimicking nanozymes capable of combating *C. albicans*, including its biofilms. In the presence of exogenous H_2_O_2_, I-CDs catalyze the generation of ROS, including •OH, which can penetrate the biofilm structure and effectively destroy fungal cells within. Increasing the iodine content of the I-CDs enhances both their peroxidase-like activity and antifungal efficacy, which is crucial for eradicating biofilms that typically exhibit high resistance to conventional treatments. Under experimental conditions (I-CDs 2.72 mg/mL, H_2_O_2_ 0.5 mM, visible light irradiation for 120 min), the system achieved over 90% inhibition of C. albicans growth. The antifungal and antibiofilm effects are attributed to a synergistic mechanism involving both the peroxidase-like activity of the nanozymes, which generates ROS that damage fungal cells, and photocatalytic activity, which further promotes ROS production under light exposure, enhancing biofilm penetration and killing efficacy. TEM characterization revealed that I-CDs-3 have a spherical morphology with a well-dispersed particle size of 2–5 nm, facilitating uniform ROS distribution throughout the biofilm. Notably, the I-CDs/H_2_O_2_ system was effective at H_2_O_2_ concentrations lower than physiological levels, highlighting its potential safety for therapeutic applications. Overall, these findings indicate that I-CDs represent a promising antibiofilm strategy against *C. albicans*, enabling the eradication of resilient biofilms [114].

Currently, the literature does not report the use of nanozymes beyond those exhibiting peroxidase-like activities in combination with photocatalytic effects. For the eradication of *Candida* biofilms, peroxidase-like nanozymes—and, potentially, oxidase-like nanozymes—are the most relevant, as other types primarily act to mitigate oxidative stress within cells, thereby counteracting the desired antifungal mechanism. Nonetheless, these alternative classes of nanozymes may hold clinical value in different therapeutic contexts, such as wound healing, where the controlled modulation and attenuation ROS could facilitate tissue regeneration and recovery. The classification of nanozymes and the action of POD-like enzymes on *Candida* biofilms are shown in Figure 5. This appears to be a very promising approach, as such a combination would allow the cumulative effects of the enzymatic activity and photocatalysis, potentially increasing the success rate of *Candida* spp. biofilm eradication. The combination of nanozymes with their photocatalytic activity represents a highly advanced and mechanistically rational strategy for the eradication of *Candida* spp. biofilms in clinical settings. *Candida* biofilms are composed of densely packed fungal cells embedded within an EPS matrix, which acts as a multifactorial barrier, reducing the diffusion of antifungal agents and protecting cells from oxidative and immune-mediated stress. As mentioned before, peroxidase-like nanozymes can catalyze the conversion of exogenous or endogenous substrates, such as hydrogen peroxide, into ROS, including •OH and O_2_•^−^. When coupled with photocatalysis, the local production of ROS is further amplified under light irradiation, allowing the oxidative species to penetrate deeper layers of the biofilm matrix and reach fungal cells that are typically protected from conventional antifungal agents. The synergistic effect of enzymatic catalysis and light-activated ROS generation enhances both the spatial distribution and temporal kinetics of oxidative stress within the biofilm, resulting in more efficient structural degradation of the EPS and eradication of embedded fungal cells [99]. Additionally, the physicochemical properties of nanozymes—including particle size, doping with heteroatoms (e.g., nitrogen, iodine, or metals), and surface functionalization—can be precisely tuned to optimize ROS yield, biofilm penetration, and selectivity toward fungal cells over mammalian tissues, minimizing cytotoxicity [115]. In medical applications, such nanozyme-photocatalyst hybrid systems could be deployed to eradicate biofilms on indwelling devices, such as catheters and prosthetic implants, which are common reservoirs of persistent and drug-resistant *Candida* infections. Overall, the integration of nanozyme catalysis with photocatalytic activation offers a potent, multifaceted antifungal strategy that exploits both enzymatic and photochemically enhanced oxidative mechanisms, providing a highly promising avenue for the prevention of *Candida* biofilm-associated infections.

### 6.5. Carbon-Based Nanomaterials: Photocatalytic Properties and Anti-Candida Biofilm Applications

Carbon-based nanomaterials, including especially CDs, are a class of nanoscale materials distinguished by high surface area, tunable surface chemistry, and exceptional electronic properties. These features make them particularly suitable for photocatalytic applications, as they can efficiently absorb light, generate ROS, and promote charge separation, thereby enhancing oxidative processes. In the context of microbial control, carbon-based photocatalysts offer the ability to disrupt cellular structures and inhibit biofilm formation under light irradiation, making them promising candidates for antifungal and antimicrobial strategies [116].

CDs exert antifungal effects against *C. albicans* biofilms primarily through photocatalytic and surface-interaction mechanisms. Under visible-light irradiation, CDs generate ROS. Additionally, their positively charged or functionalized surfaces enhance adhesion to negatively charged fungal cells, promoting internalization and disrupting cellular integrity. This combined action inhibits biofilm formation, reduces cell viability within established biofilms, and prevents the maturation of biofilm architecture, making CDs a potent nanomaterial for antifungal applications [40]. The mechanisms of combating *Candida* spp. and its biofilm using CDs are presented in Figure 6.

Gao et al. [117] reported the synthesis of nitrogen- and chlorine-co-doped carbon dots (N/Cl-CDs) through a one-step hydrothermal process employing a choline chloride–urea deep eutectic solvent. The resulting nanomaterials demonstrated excellent photocatalytic antifungal properties against *C. albicans*. Under visible-light irradiation, N/Cl-CDs achieved complete inhibition of planktonic yeast growth at a concentration of 7 mg/mL within 80 min. Microscopic observations confirmed extensive structural damage to fungal cells, with envelopes displaying deformation, distortion, and wrinkling. Beyond the direct fungicidal effect, the disruption of cellular integrity and metabolic activity significantly reduced the ability of *C. albicans* to establish biofilms, a process central to pathogenic persistence and therapeutic resistance. Taken together, these results underscore the potential of N/Cl-CDs as multifunctional nanomaterials capable of both eradicating fungal cells and suppressing biofilm development, offering a promising platform for future antifungal strategies.

In the study by Cong et al. [118], a BiOBr@CDs-TiO_2_−x heterojunction with a double Z-scheme architecture was constructed, comprising defective BiOBr microspheres with dominant (110) facets and a TiO_2_−x shell functionalized with CDs. The amine-functionalized CDs played a dual role by simultaneously reducing and doping TiO_2_−x, generating oxygen vacancies and increasing the positive surface charge, which facilitated interactions with microbial cells. Photocatalytic experiments were conducted under visible-light irradiation using *C. albicans* biofilms at a concentration of 1 × 10^8^ CFU/mL. The heterojunction exploited the double Z-scheme mechanism to promote efficient spatial charge separation, thereby enhancing ROS generation and minimizing electron–hole recombination. The BiOBr@CDs-TiO_2_−x nanostructures achieved 99.9% eradication of *C. albicans* cells within the biofilm in 1 h. Morphological analyses revealed extensive damage to cell structure and biofilm integrity, indicating synergistic mechanical and photocatalytic disruption. Moreover, the system concurrently promoted the degradation of organic pollutants, highlighting its multifunctional potential for antimicrobial applications and use in medical environments.

As described in the previous section, Li et al. also demonstrated the application of carbon dots and their photocatalytic mechanism. The study employed iodine-doped carbon dots (I-CDs), with I-CDs-3 selected for detailed characterization due to its superior peroxidase activity.

Darbari et al. [119] evaluated the photocatalytic antifungal activity of TiO_2_/branched CNT nanostructures against *C. albicans* (ATCC 10231) biofilms. Branched CNTs were synthesized via plasma-enhanced chemical vapor deposition on a silicon substrate using nickel as a catalyst, followed by deposition of TiO_2_ nanoparticles and annealing at 500 °C. Biofilms were formed in YNB medium with 50 mM glucose at 37 °C for 48 h and irradiated with visible light (halogen lamp, 100 mW/cm^2^). Cell viability (MTT assay) and morphology (SEM) showed that TiO_2_/branched CNT significantly inhibited biofilm development compared to silicon, TiO_2_ thin films, and TiO_2_/CNTs. SEM revealed yeast cells enmeshed by branched CNTs with surface defects, indicating mechanical and photocatalytic disruption. Enhanced antifungal activity was attributed to visible-light-induced electron–hole pair generation, low recombination, and high surface area. The study confirms that TiO_2_/branched CNTs are an effective photocatalytic nanomaterial for controlling and eradicating *C. albicans* biofilms.

Carbon-based nanomaterials, including CDs and carbon nanotubes, represent a highly promising class of photocatalytic agents for the disruption of *C. albicans* biofilms. Their exceptional physicochemical properties—high surface area, tunable surface functionality, and efficient generation of ROS under visible-light irradiation—enable both oxidative and mechanical damage to fungal cells and biofilm matrices. Experimental evidence indicates that functionalized or doped CDs, as well as TiO_2_-decorated branched carbon nanotubes, can achieve substantial antifungal efficacy, markedly reducing cell viability and inhibiting biofilm development. Nonetheless, comprehensive investigations are required to elucidate their biocompatibility, pharmacokinetics, and translational potential within clinical and medical settings, thereby supporting their integration as advanced antifungal therapeutics.

### 6.6. Modified Composites and Medical Biomaterials in the Control of Candida spp.

Thanks to the successful outcomes of the aforementioned photocatalytic methods, many studies have focused on incorporating these mechanisms into medical biomaterials and composites to address the problem of *Candida* spp. biofilm formation and its potential consequences for patient groups prone to such complications. However, practical applications to date have primarily focused on metal and metal oxide nanoparticles, with TiO_2_ being among the most commonly used nanoparticles with catalytic properties. Most studies have focused on *C. albicans*, although some novel methods have also been applied to *C. glabrata*. It is anticipated that the nanomaterials described in the previous section may soon find practical application in healthcare settings for the eradication of *Candida* spp. biofilms from medical devices and surfaces.

Appropriate coatings on surfaces and medical equipment are fundamental for infection control. The use of light to enhance the photocatalytic activity of such coatings represents a promising approach in modern medicine. Various nanomaterials can be incorporated into medical coatings to prevent microbial growth and biofilm formation. Metallic nanoparticles provide strong antimicrobial activity, while metal oxide nanoparticles, such as TiO_2_ and ZnO, offer photocatalytic properties that increase their effectiveness under light. Carbon-based nanostructures can further improve surface durability while supporting antimicrobial or photocatalytic functions.

Ballo et al. [120] investigated the potential application of photocatalysis to achieve antimicrobial effects against *Candida* spp. The study focused on the antifungal properties of copper (Cu) and Cu-alloy surfaces. Using Direct Current Magnetron Sputtering (DCMS), copper nanoparticle (Cu-NP) coatings were deposited onto various substrates. One such modified material, copper-sputtered polyester (Cu-PES), was evaluated in this work. Previous studies have indicated that this material is safe for both healthcare personnel and patients, as metallic copper is considered a rare sensitizing agent and has not been associated with skin irritation. Furthermore, copper oxides on PES exhibit a lower band gap (1.9 eV) than TiO_2_, which enables absorption of visible light and provides superior performance. To prepare the modified polyester, copper was deposited onto the fabric surface using DCMS at temperatures below 130 °C. This process produced CuO and Cu_2_O layers on the polyester, with their composition verified by X-ray fluorescence and their optical absorption properties analyzed by diffuse reflectance spectroscopy. The antifungal activity of Cu-PES was subsequently tested under both dark conditions and actinic light. Cu-PES samples were placed directly onto plates containing *C. albicans* and *C. glabrata*. Actinic light was generated using tubular lamps with a visible emission spectrum of 400–700 nm and an intensity of 4.65 mW/cm^2^. Fungicidal activity was observed for Cu-PES in both dark and illuminated conditions, though more rapid effects were achieved under visible light, with a reduction of 3.4 log_10_ CFU after 30 min and 3.9 log_10_ CFU after 60 min. In contrast, control experiments with uncoated PES showed survival of the *Candida* cells in both conditions, confirming that eradication was due to the photocatalytic activity of the copper coating.

Grijalva-Castillo et al. [82] investigated the antifungal activity of SnO_2_ thin films doped with silver and cuprite nanoparticles, as well as WO_3_ thin films, activated by ultraviolet C (UVC) radiation. SnO_2_ films were prepared by spray pyrolysis of a stannous chloride solution onto heated glass substrates, followed by doping with Cu_2_O or Ag nanoparticles via drop-casting. WO_3_ films were fabricated by reactive DC sputtering of a high-purity tungsten trioxide target onto glass substrates under an Ar/O_2_ atmosphere, with film thickness controlled by deposition time and post-deposition annealing at 500 °C to improve crystallinity. The antifungal activity of the films was evaluated against *C. albicans*. For SnO_2_, films were irradiated with 254 nm UVC light for 5 min, and then both the irradiated films and control glass surfaces were incubated in Sabouraud Dextrose Broth at 37 °C for 24 h. WO_3_ films were inoculated with 10 μL of microbial suspension, air-dried at room temperature for 24 h, and subsequently exposed to 254 nm UVC light, with glass surfaces serving as controls. SnO_2_ films doped with Cu_2_O or Ag nanoparticles exhibited strong antifungal activity, achieving over 80% inhibition even without additional treatment. UVC irradiation further enhanced efficacy, resulting in complete suppression of fungal growth. Morphological analysis revealed significant cell damage, particularly in Ag-doped films, and biofilm formation was effectively prevented by UVC exposure. WO_3_ films showed a thickness-dependent antifungal effect, with thicker films producing greater inhibition. Films not exposed to UVC slightly promoted *C. albicans* growth, while 200 nm films reduced cell viability to 13.8%, demonstrating the strongest effect.

Faudoa-Arzate et al. [121] investigated the photocatalytic activity of SnO_2_ thin films to enhance disinfection under UV light, highlighting their potential application on medical devices, walls, floors, and other hospital surfaces. The SnO_2_ films were prepared via spray pyrolysis, where a 0.5 M SnCl_2_ solution in distilled water was nebulized onto a glass substrate heated to 400 °C, forming the thin film. Antifungal activity was tested against *Candida albicans*, comparing standard UV light treatment with the combination of UV light and SnO_2_ coatings. Experiments were conducted on four groups: slides without SnO_2_ films not exposed to UV light (Control 1); slides with SnO_2_ films not exposed to UV light (Control 2); slides without SnO_2_ films exposed to UV light; and slides with SnO_2_ films exposed to UV light. The results demonstrated that the combined application of UV light on SnO_2_-coated slides significantly enhanced disinfection. *C. albicans* exposed to UV light alone exhibited inhibition rates of 7% after 5 min, 28% after 10 min, and 37% after 15 min, while when deposited on SnO_2_ films and irradiated, inhibition rates increased markedly to 40%, 52%, and 60% at the corresponding time points.

The development of *Candida* spp. biofilms also poses a significant challenge for individuals with facial and maxillofacial prostheses. These devices are commonly provided to patients who have experienced traumatic injuries, undergone extensive surgical procedures resulting in deformities, or present with congenital craniofacial anomalies. Their use helps restore facial appearance, improve speech and masticatory functions, promote tissue healing, and alleviate psychological distress. However, these prostheses are typically made from materials such as silicone elastomers, which are prone to microbial colonization, including by *C. albicans*. Colonization by *C. albicans* leads to biofilm formation, accelerating material degradation, causing skin irritation, and compromising prosthesis function, with the prosthetic surface particularly favoring yeast overgrowth [122].

It has been proposed to incorporate the photocatalytic properties of TiO_2_ nanoparticles (NPs) into the materials used for prostheses, in order to achieve antimicrobial effects against *C. albicans* upon irradiation. Widodo et al. [123] conducted research on modified polyurethane, a highly biocompatible material commonly used for prosthesis fabrication but prone to *C. albicans* colonization. Polyurethane plate samples were prepared with and without TiO_2_ as a filler at concentrations of 1%, 2%, 3%, and 4%. Silanized TiO_2_ was first mixed with 12 g of polyurethane Part B and stirred for 2 min, then combined with 12 g of Part A (1:1 ratio) and stirred again until homogeneous. The mixture was degassed in a vacuum chamber to remove air bubbles, poured into molds, and allowed to cure. For surface-coated samples, polyurethane plates were initially prepared without TiO_2_. Their surfaces were treated with a silane coupling agent and left to dry. Silanized TiO_2_ was then mixed with ethanol to form a paste at the target concentration and stirred for 30 min. The paste was evenly applied to the surface using a spatula via the slip-casting method. All prepared samples, whether containing TiO_2_ as a filler or a surface coating, were irradiated with UV light (366 nm, 15 cm, 1 h) to activate the photocatalytic properties. After contamination with *C. albicans* and incubation for 48 h, the addition of TiO_2_ at concentrations of 1–4% significantly reduced fungal colony formation. The strongest antifungal effect was observed with 4% TiO_2_ applied as a surface coating.

Another similar study by El Shafie et al. [124] investigated the photocatalytic effect of TiO_2_ nanoparticles in inhibiting *C. albicans* biofilm formation on silicone elastomer facial prostheses used in patients with facial defects. In this study, TiO_2_ nanoparticles were incorporated into the silicone elastomer at varying concentrations and tested under UV-A light exposure. The results demonstrated significant antifungal activity, with the best outcome observed at 6% TiO_2_ loading. The authors concluded that patients could activate the embedded nano-TiO_2_ by brief exposure to natural UV-A sunlight (3–5 min during specific daytime hours) or by using UV-A lamps (approximately 2 h at night) to achieve optimal antifungal effects. Both studies highlight a promising strategy to reduce microbial colonization and inhibit biofilm formation on facial/maxillofacial prostheses, potentially decreasing infection risk and hospitalization rates among these patients.

*Candida* spp. often form biofilms on oral prostheses. One of the most common manifestations of oral candidiasis is denture-associated stomatitis, which results from *Candida* biofilm formation on the prosthesis surface. This complication is particularly prevalent in the elderly population and, according to studies, may affect up to 70% of denture wearers [125].

Liu et al. [126] proposed a potential solution for recurrent denture stomatitis by developing a biomaterial with photocatalytic antifungal properties. The photocatalytic activity was achieved by incorporating TiO_2_-HAP (hydroxyapatite, HAP) nanoparticles into self-curing denture resin. The addition of HAP enhances the light absorption efficiency of TiO_2_ nanoparticles in the visible range (400–700 nm) compared with conventional TiO_2_ NPs. PMMA-based dental resin specimens containing TiO_2_-HAP were prepared for testing. TiO_2_-HAP was incorporated into the base powder of a self-curing denture resin at various ratios (0, 0.5, 1, 1.5, and 2 parts per 100 parts powder) using a ball mill to ensure thorough mixing. The powder was then combined with the liquid component according to a 2:1 powder-to-liquid ratio. The mixture was placed into silicone molds and allowed to set for 30 min. After curing, the specimens were polished to achieve flat and smooth surfaces, then immersed in aqueous ethanol, rinsed with sterile PBS, and air-dried before 24-h UV irradiation for sterilization. Before assessing antifungal activity, the cytotoxicity of the modified resin was evaluated on gingival fibroblasts, showing no significant signs of toxicity. Its antifungal efficacy against *C. albicans* was then assessed using a biofilm accumulation assay. Specimens containing TiO_2_-HAP under visible light demonstrated 94% antifungal activity. SEM images further confirmed these findings, showing substantial differences in *C. albicans* growth on modified versus unmodified PMMA. Conventional PMMA exhibited extensive fungal growth, whereas TiO_2_-HAP-modified PMMA under visible light showed only minimal *C. albicans* adhesion. At a low concentration of 1 wt% TiO_2_-HAP, the modified PMMA displayed low cytotoxicity and strong visible-light-induced photocatalytic antifungal properties. Under visible light, TiO_2_-HAP generates ROS that damage *C. albicans* cells and prevent their transition from yeast to filamentous forms, thereby inhibiting biofilm formation on the PMMA surface. These results suggest a promising approach to preventing *Candida* biofilm formation on dental prostheses and reducing denture-associated stomatitis in elderly patients.

A similar idea presented by Kim et al. [110] was presented in the previous part of our article in the section about zinc oxide (ZnO) nanoparticles. This study proposed a similar approach, but the PMMA was incorporated with ZnO nanoflakes instead of TiO_2_-Hap NPs.

Another area where the photocatalytic properties of nanomaterials have been explored is orthodontics. Orthodontic therapy involving brackets is a common and effective method for treating malocclusion. However, this long-term treatment often compromises oral hygiene, creating favorable conditions for biofilm formation and *C. albicans* colonization.

In a study by Alhasani et al. [127], 13.8% of orthodontic patients were identified as oral carriers of *C. albicans*, and among 40 isolates, 21 exhibited biofilm-forming capacity. Similarly, Grzegocka et al. [128] demonstrated that orthodontic appliances promote *Candida* colonization, although in their study the strains did not display enhanced biofilm activity. Both reports highlight that orthodontic appliances increase the risk of oral candidiasis.

To address *Candida* colonization during orthodontic treatment, the photocatalytic properties of TiO_2_ nanoparticles have been proposed for application on stainless steel brackets. Cao et al. [129] coated brackets with thin TiO_2_ films using radio frequency (RF) magnetron sputtering. Both undoped and nitrogen-doped TiO_2_ films were deposited on stainless steel substrates under controlled conditions (200 W, 300 °C, 1.0 Pa, 99.99% pure target). Nitrogen doping was employed to extend TiO_2_ absorption into the visible-light range. After deposition, films were cooled and annealed in nitrogen at various temperatures. Morphological analysis revealed uniform, compact, fine-grained surfaces in N-doped films, while undoped TiO_2_ exhibited polyhedral grains with smooth facets. Antimicrobial testing against *C. albicans* demonstrated that TiO_2_-coated brackets inhibited fungal growth. The highest photocatalytic and antimicrobial activity under visible light was achieved with the following parameters: sputtering temperature of 300 °C, sputtering time of 180 min, Ar/N ratio of 30:1, and annealing at 450 °C.

Özyıldız et al. [130] modified polycrystalline alumina ceramic Intrigue brackets (Lancer Orthodontics, Mexicali, Mexico) by applying a thin film of TiO_2_ using the sol–gel dip-coating method. Titanium (IV) isopropoxide was mixed with 2-propanol, hydrochloric acid, and 3 wt% polyethylene glycol (PEG) to improve adhesion and reduce cracking. The solution was stirred at room temperature for 3 h and subsequently aged at 4 °C for 24 h. The brackets were ultrasonically cleaned, dip-coated at a controlled speed, and dried sequentially at room temperature and then at 120 °C. This coating process was repeated three times to increase film thickness. Finally, the coated brackets were calcined at 500 °C to obtain the final TiO_2_ layer. After preparation, the photocatalytic antimicrobial effects were tested against C. albicans. Both coated and uncoated brackets were placed in sterile Petri dishes containing 1 mL of a *C. albicans* suspension. Specimens were then exposed to UVA light (1.0 mW/cm^2^) for 1 h, while a separate group was kept under dark conditions. Results showed that UVA-irradiated coated brackets achieved 93% inhibition of *C. albicans*, whereas uncoated brackets reached only 16% inhibition under the same conditions. In contrast, coated brackets kept in the dark exhibited only 12.5% inhibition. These findings confirm that the antimicrobial mechanism is based on the photoinduced oxidative activity of TiO_2_, which leads to the generation of ROS and subsequent fungal cell death.

This subsection illustrates how the photocatalytic properties of nanomaterials show significant potential in preventing and eradicating *Candida* spp. biofilms in the medical field. Current research has primarily focused on incorporating particularly metal and metal oxides into coatings and a variety of materials used in healthcare settings, including prosthetic devices, dental resins, and orthodontic appliances, demonstrating strong antifungal effects upon light activation. These modifications not only reduce microbial colonization and inhibit biofilm formation but also help maintain the structural integrity and functionality of medical devices, thereby improving their longevity and safety. By limiting *Candida* overgrowth, photocatalytic nanomaterials contribute to the prevention of *Candida* infections, which is particularly important for patients requiring long-term protection against *Candida* infection. Overall, the application of photocatalytic nanomaterials represents a valuable strategy for enhancing infection control, offering sustained anti-biofilm activity against *Candida* and providing significant benefits for vulnerable patient populations in clinical environments.

## 7. Materials and Methods

We conducted a systematic literature search to investigate the application of nanomaterials in photocatalysis for the eradication of *Candida* spp. and their biofilms. The search strategy employed specific keywords, including “nanomaterials in photocatalysis against *Candida* biofilms,” “nanoparticles photocatalytic activity against *Candida* biofilm,” “*Candida* spp. biofilm eradication by carbon dots,” “nanozymes as photocatalysts against *Candida*,” and “nanomaterial-based photocatalytic coatings for *Candida* spp. biofilm eradication.”

Our analysis encompassed the most relevant nanomaterials reported to exhibit activity against *Candida* spp. biofilms, along with concise descriptions of their characteristics, the mechanisms of photocatalysis in disrupting *Candida* biofilms, and—most importantly—their potential applications in healthcare environments to address the growing challenge of *Candida* spp. biofilm-associated infections.

We focused primarily on original research articles to obtain a comprehensive and practice-oriented overview of both the efficacy and practical applicability of these methods. Particular emphasis was placed on studies published in recent years to ensure that our findings reflect the most current progress in this rapidly developing field. This approach was motivated by the urgent need to highlight potential future directions in light of the alarming increase in *Candida* infections, including those caused by drug-resistant strains, within healthcare settings.

To facilitate understanding, the collected information was summarized in schematic illustrations that highlight the key relationships and mechanisms underlying the described topics.

## 8. Conclusions

In this review, we present a comprehensive overview of recent advances and potential future directions in the application of nanomaterials in photocatalytic processes for the eradication of *Candida* spp. and their biofilms.

We emphasized the growing clinical problem of *Candida* spp. biofilms in healthcare environments and the associated risks linked to the rising incidence of infections that are increasingly difficult to treat.

The stages of *Candida* biofilm development and their key features, which hinder effective eradication, were outlined.

We also summarized the most effective nanomaterials reported to date, providing a brief characterization of their properties.

Furthermore, we described the mechanisms through which photocatalysis acts on *Candida* biofilms and demonstrated its potential as a valuable alternative in light of the limitations of conventional antifungal drugs.

The central aim of this review was to illustrate how nanomaterials can be exploited as photocatalysts for the eradication of *Candida* spp. biofilms. We discussed in detail the mechanisms of action, experimental conditions, and reported efficacy of nanomaterials studied in recent years, including metals, titanium dioxide, zinc oxide, transition-metal oxides, carbon dots, and nanozymes. These represent a promising class of materials with potential for broader application in healthcare environments; however, most require further large-scale studies and translational strategies to enable practical implementation in clinical settings.

Finally, we reviewed existing applications of photocatalytic nanomaterials in coatings and medical devices, which have already demonstrated the ability to prevent and eradicate *Candida* spp. biofilms. We anticipate that the other nanomaterials discussed in our review will soon be tested in similar biomedical contexts, contributing additional strategies to address the growing challenge of *Candida* infections in medical environments. In future perspectives, special attention should be given to the development of intelligent nanomaterials capable of responding to specific biological stimuli, such as pH or enzymatic activity, to enable targeted and on-demand antimicrobial action. Moreover, integrating photocatalytic approaches with photodynamic therapy (photocatalytic–photodynamic combined therapy) may provide synergistic therapeutic effects through enhanced ROS generation and selective microbial inactivation. Such multifunctional systems hold significant promise for next-generation antifungal treatments and the design of self-sterilizing biomedical surfaces.

## Figures and Tables

**Figure 1 molecules-30-04500-f001:**
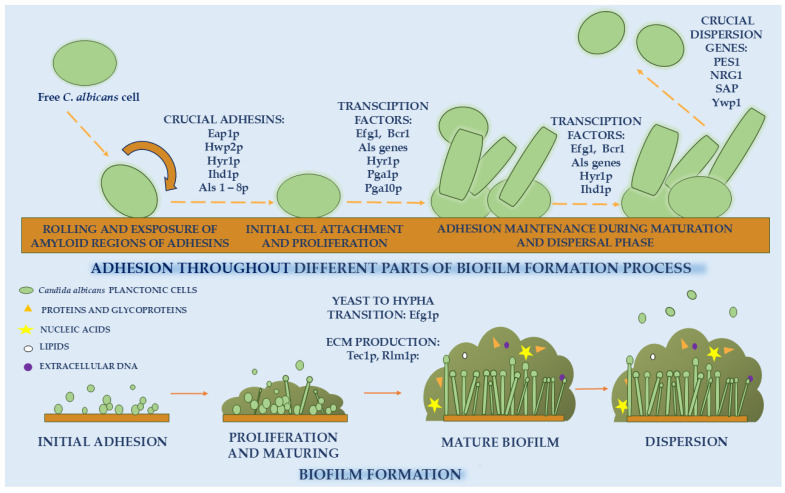
Molecular and cellular mechanisms of *Candida albicans* biofilm formation, showing sequential stages from adhesion, proliferation, and maturation to dispersion. The figure highlights the roles of key adhesins (Eap1p, Hwp2p, Hyr1p, Ihd1p, Als proteins) and transcription factors (Efg1, Bcr1, Tec1, Rlm1)—throughout different parts of the biofilm formation process and extracellular matrix (ECM) components—including proteins, nucleic acids, lipids, and extracellular DNA—in biofilm regulation and structural development.

**Figure 2 molecules-30-04500-f002:**
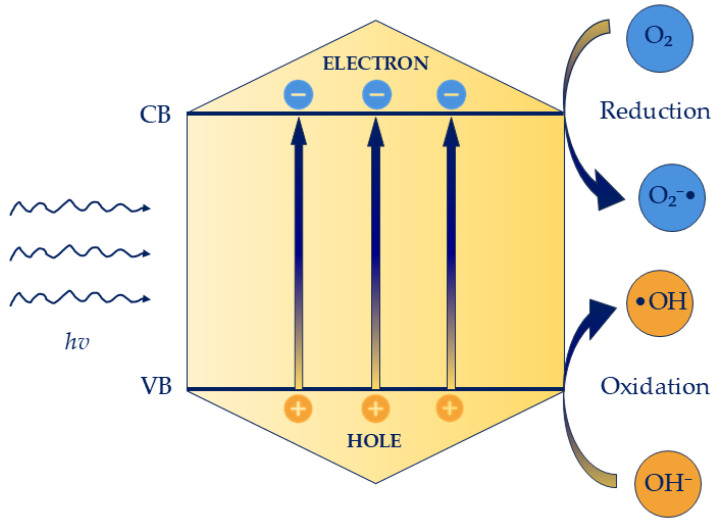
The basic mechanism of photocatalysis. Upon light irradiation with photon energy equal to or greater than the band gap of the photocatalyst, electrons (e^−^) in the valence band (VB) are excited to the conduction band (CB), leaving behind positively charged holes (h^+^) in the VB. The photogenerated charge carriers subsequently migrate toward the catalyst surface, where they participate in redox reactions. Electrons (e^−^) in the CB reduce molecular oxygen (O_2_) to superoxide anions (O_2_•^−^) while holes (h^+^) in the VB oxidize hydroxide ions (OH^−^) to generate hydroxyl radicals (•OH). The overall photocatalytic efficiency depends on the balance between light absorption, charge carrier separation, and interfacial redox dynamics.

**Figure 3 molecules-30-04500-f003:**
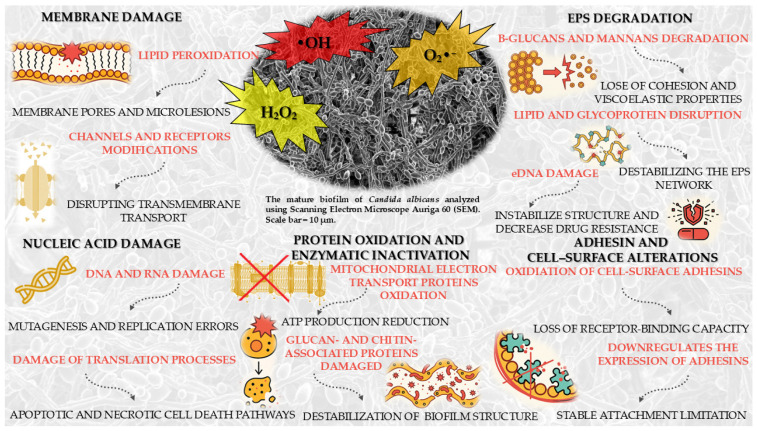
Effects of photocatalysis on *Candida* spp. biofilms: mechanisms of ROS-induced damage to membranes, nucleic acids, proteins, enzymes, extracellular polymeric substances (EPS), and cell-surface adhesins.

**Figure 4 molecules-30-04500-f004:**
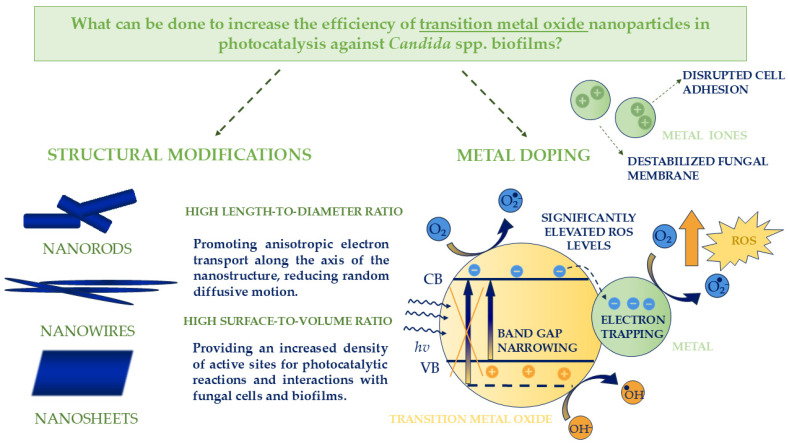
Strategies to enhance the effectiveness of combating *Candida* spp. biofilms using transition metal oxides as photocatalysts: structural modification and metal doping. Structural engineering approaches, such as the formation of nanorods, nanowires, or nanosheets, improve charge carrier mobility and enhance the surface-to-volume ratio, thereby increasing the number of active sites available for photocatalytic reactions and interactions with fungal biofilm components. Metal doping narrows the band gap and introduces intermediate energy levels that promote efficient photon absorption and electron trapping, resulting in elevated ROS generation. The released metal ions can further destabilize the fungal cell membrane and disrupt cell adhesion, amplifying antifungal efficacy. Together, these modifications improve electron–hole separation, boost ROS yield, and enhance biofilm penetration, offering a rational strategy to increase the antifungal performance of transition metal oxide-based photocatalysts in clinical and environmental applications.

**Figure 5 molecules-30-04500-f005:**
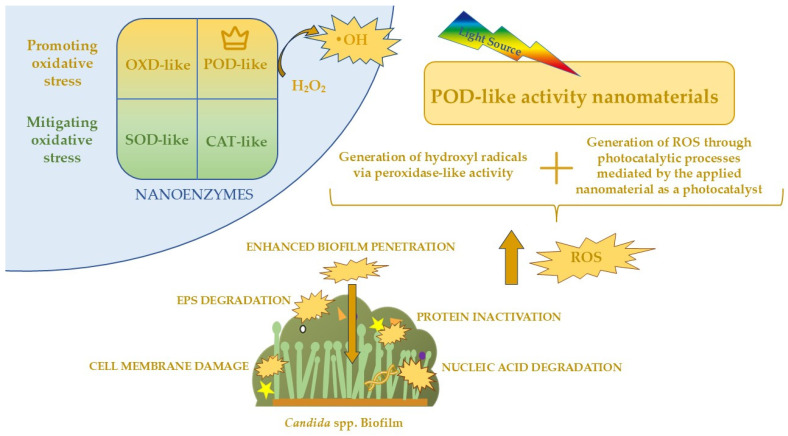
Types of nanozymes by ROS modulation and POD-like nanozyme action on *Candida* biofilms. Nanozymes with POD- and oxidase (OXD)-like activities promote oxidative stress by catalyzing the decomposition of hydrogen peroxide (H_2_O_2_) or oxygen (O_2_) into highly reactive species such as hydroxyl radicals (•OH) and superoxide anions (O_2_•^−^), while catalase (CAT)- and superoxide dismutase (SOD)-like nanozymes primarily mitigate oxidative stress by decomposing excess ROS. Under light irradiation, POD-like nanomaterials additionally exhibit photocatalytic properties, enhancing ROS generation through synergistic photochemical and enzymatic processes. The resulting oxidative species penetrate the extracellular polymeric substance (EPS) matrix of *Candida* biofilms, leading to EPS degradation, protein and nucleic acid oxidation, and cell membrane disruption. This dual-function mechanism—combining peroxidase-like catalysis with photocatalysis—facilitates deeper ROS diffusion and more efficient eradication of fungal biofilms, offering a promising approach for the treatment of biofilm-associated *Candida* infections.

**Figure 6 molecules-30-04500-f006:**
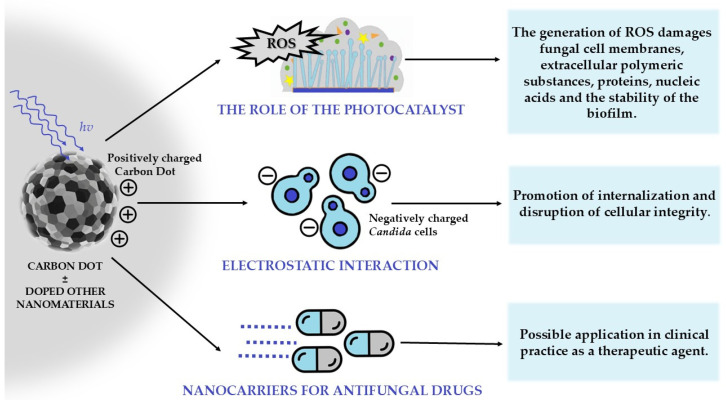
Mechanisms of combating *Candida* spp. and its biofilm using carbon dots (CDs): photocatalysis, electrostatic interactions, and drug delivery systems.

## Data Availability

No new data were created or analyzed in this study. Data sharing is not applicable to this article.

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
