# Peer review of "Nanomaterials for Photocatalytic Inactivation and Eradication of Candida spp. Biofilms in Healthcare Environment: A Novel Approach in Modern Clinical Practice"

_molecules, 2025, doi:10.3390/molecules30234500_

Round 1

Reviewer 1 Report

Comments and Suggestions for Authors

This review explores the potential application of nanomaterials in photocatalytic inactivation and eradication of Candida biofilms. The article has a clear structure and comprehensive content, covering the formation mechanism of Candida biofilm, drug resistance mechanism, classification and photocatalytic mechanism of nanomaterials, as well as practical application cases of various nanomaterials in medical environments. However, the following aspects still need to be considered:

  1. The article cites a large number of studies, but does not evaluate the reliability of research methods, sample size, and experimental design. Please increase the quality assessment of cited literature, such as using PRISMA guidelines or similar tools, to enhance the rigor of the review.
  2. There are significant differences in conditions such as light intensity, wavelength, and nanomaterial concentration among different studies, making it difficult to compare horizontally. It is recommended to list the key experimental parameters in a table or appendix for readers to evaluate and repeat the experiment.
  3. There is insufficient discussion on the toxicity and biocompatibility of nanomaterials. Although the cytotoxicity of some materials is mentioned, there is a lack of systematic evaluation and clinical safety analysis.Please add a section specifically discussing the biocompatibility, in vivo toxicity, and potential risks of nanomaterials.
  4. The paper did not discuss the potential adaptation or resistance mechanisms of Candida to photocatalytic ROS production. Please explore potential drug resistance mechanisms and propose a nano material strategy for “anti drug resistance design”.
  5. Although clinical applications are mentioned, the key steps and challenges from laboratory to clinical are not specifically explained. Please add a section on “Clinical Translation Challenges and Strategies” to discuss issues such as regulation, cost, and large-scale production.
  6. generate reactive oxygen species (ROS) upon irradiation (line 329) need support. For example, Title: Synergistic electric fields induced by unilateral doping modulation for enhanced organic pollutant degradation and sterilization.
  7. The mechanism diagram is too simplified. Partial mechanism diagrams (such as Fig.2 and Fig.5) fail to fully demonstrate the specific migration pathways of electron hole pairs during the photocatalytic process. Please add more detailed mechanism explanations in the caption.
  8. The paper mainly focuses on C. albicans, with insufficient discussion on emerging drug-resistant strains such as C. auris. It is better to expand the research review on atypical bacterial species such as C. auris and C. lusitaniae.
  9. photocatalytic mechanism in line 370 need support. Title: Interfacial electric field steering S-scheme charge transfer in MIL-88A(Fe)/polydopamine Heterojunctions: Dual-redox pathways for efficient pollutant mineralization.
  10. The advantages and disadvantages of various nanomaterials are scattered throughout the text, lacking horizontal comparison. Please add a comprehensive comparison table that covers dimensions such as photocatalytic efficiency, stability, toxicity, and cost.
  11. It is better to add prospects for cutting-edge directions such as intelligent nanomaterials and photocatalytic photodynamic combined therapy in the conclusion section.

Author Response

This review explores the potential application of nanomaterials in photocatalytic inactivation and eradication of Candida biofilms. The article has a clear structure and comprehensive content, covering the formation mechanism of Candida biofilm, drug resistance mechanism, classification and photocatalytic mechanism of nanomaterials, as well as practical application cases of various nanomaterials in medical environments. However, the following aspects still need to be considered:

We sincerely thank you for taking the time to provide a thorough and insightful evaluation of our manuscript. We are truly grateful for your constructive comments and valuable suggestions, which will undoubtedly help us improve the quality and rigor of our work.

  1. The article cites a large number of studies, but does not evaluate the reliability of research methods, sample size, and experimental design. Please increase the quality assessment of cited literature, such as using PRISMA guidelines or similar tools, to enhance the rigor of the review.

The aim of our work was to construct a narrative review that highlights a novel topic from multiple perspectives. Since many aspects of this field are quite new, it was important for us to include a wide range of studies to provide a comprehensive overview. While we acknowledge the value of formal quality assessment tools such as PRISMA, we chose not to apply them because our review is intended as a narrative, rather than a systematic, review. Our goal was to provide a broad and integrative discussion of emerging research directions rather than a strict meta-analysis, which allows us to include diverse studies, including preliminary and exploratory reports that are essential to convey the current state of this rapidly evolving field.

  1. There are significant differences in conditions such as light intensity, wavelength, and nanomaterial concentration among different studies, making it difficult to compare horizontally. It is recommended to list the key experimental parameters in a table or appendix for readers to evaluate and repeat the experiment.
  2. There is insufficient discussion on the toxicity and biocompatibility of nanomaterials. Although the cytotoxicity of some materials is mentioned, there is a lack of systematic evaluation and clinical safety analysis.Please add a section specifically discussing the biocompatibility, in vivo toxicity, and potential risks of nanomaterials.
  1. The advantages and disadvantages of various nanomaterials are scattered throughout the text, lacking horizontal comparison. Please add a comprehensive comparison table that covers dimensions such as photocatalytic efficiency, stability, toxicity, and cost.

Regarding the reviewer’s comments on experimental conditions, toxicity, and biocompatibility of nanomaterials, we acknowledge these important aspects. In our manuscript, we have addressed them wherever possible within the relevant subsections, presenting information as reported in the cited studies. While the reviewer suggests including a table summarizing key experimental parameters and nanomaterial characteristics, we believe that doing so would significantly broaden the scope of the work, which is already extensive, and might not fully capture the context-dependent variations, such as differences in preparation conditions or experimental settings. Therefore, we preferred to discuss these aspects within the respective subsections, alongside the cited studies, to preserve the nuance and specificity of the reported results.

  1. The paper did not discuss the potential adaptation or resistance mechanisms of Candida to photocatalytic ROS production. Please explore potential drug resistance mechanisms and propose a nano material strategy for “anti drug resistance design”.

In the revised version of our manuscript, we have added a dedicated subsection addressing the response mechanisms of Candida to oxidative stress, as suggested by the reviewer. We appreciate this valuable recommendation, as it strengthens the manuscript and provides important context for potential drug resistance mechanisms when implementing photocatalytic methods of eradication.

  1. Although clinical applications are mentioned, the key steps and challenges from laboratory to clinical are not specifically explained. Please add a section on “Clinical Translation Challenges and Strategies” to discuss issues such as regulation, cost, and large-scale production.

We appreciate the reviewer’s suggestion regarding the inclusion of a section on “Clinical Translation Challenges and Strategies.” However, given the already extensive scope and length of our manuscript, we opted to focus on the fundamental mechanisms, applications, and properties of nanomaterials in Candida biofilm inactivation. While clinical translation is an important aspect, we believe that a detailed discussion on regulatory, cost, and large-scale production issues would extend beyond the intended focus of this review.

  1. generate reactive oxygen species (ROS) upon irradiation (line 329) need support. For example, Title: Synergistic electric fields induced by unilateral doping modulation for enhanced organic pollutant degradation and sterilization.

In the revised version of our manuscript, we have updated and corrected the statement regarding the generation of reactive oxygen species (ROS) upon irradiation, as suggested by the reviewer, and have provided supporting reference with the recommended study.

  1. The mechanism diagram is too simplified. Partial mechanism diagrams (such as Fig.2 and Fig.5) fail to fully demonstrate the specific migration pathways of electron hole pairs during the photocatalytic process. Please add more detailed mechanism explanations in the caption.

We thank the reviewer for this valuable suggestion. When creating the illustrations, our goal was to present the mechanisms as clearly and transparently as possible, avoiding excessive complexity in the diagrams. To address the reviewer’s concern, we have revised and enhanced the captions of the figures to provide more detailed explanations of the photocatalytic electron–hole pair migration pathways, while keeping the diagrams themselves clear and easy to interpret.

  1. The paper mainly focuses on C. albicans, with insufficient discussion on emerging drug-resistant strains such as C. auris. It is better to expand the research review on atypical bacterial species such as C. auris and C. lusitaniae.

We appreciate the reviewer’s suggestion. However, the current literature on this topic is still very limited regarding emerging drug-resistant species such as C. auris and C. lusitaniae. This is the main reason why our manuscript primarily focuses on C. albicans, which remains the most extensively studied species in this field. Where available, we have also included studies on C. auris, C. parapsilosis, and C. tropicalis to provide a broader perspective, but unfortunately, there are simply not enough published works on those less-studied species.

  1. photocatalytic mechanism in line 370 need support. Title: Interfacial electric field steering S-scheme charge transfer in MIL-88A(Fe)/polydopamine Heterojunctions: Dual-redox pathways for efficient pollutant mineralization.

We have revised this section to support the described photocatalytic mechanism as it was suggested.

  1. It is better to add prospects for cutting-edge directions such as intelligent nanomaterials and photocatalytic photodynamic combined therapy in the conclusion section.

We thank the reviewer for highlighting these important aspects. The Conclusions section has been revised to include prospects for cutting-edge directions, and this topic has also been mentioned it Section 3 of the manuscript.

Thank you in advance!

Regards,

prof. Emil Paluch

Reviewer 2 Report

Comments and Suggestions for Authors

See attached file

Author Response

We would like to express our sincere gratitude for the constructive and insightful review. The valuable comments provided have significantly contributed to the improvement of our manuscript. In response to the suggestions, we have expanded the Conclusions section. However, we decided to retain the Materials and Methods section, as it is part of the Molecules  journal template and, in our opinion, appropriately summarizes the approach used to analyze the content presented in the study. All missing citations have been corrected. We greatly appreciate the detailed feedback, which has enabled us to further enhance the quality and clarity of the manuscript. A detailed description of the revisions is provided below:

Additional comments:

Are the figures based on literature? Add references  - The figures correspond to the content of the respective subsections; therefore, the citations included in the text refer to the graphics created for and assigned to the content of each subsection.

‘Candida spp. are estimated to cause approximately 700,000 cases of invasive candidiasis

globally each year’ ‘Nowadays, Candida auris is a major concern in critical care, with

candidiasis as the leading fungal infection, contributing—along with other fungi—to an

estimated 13 million cases and 1.5 million deaths annually worldwide’. These two sentences seem to contradict each other. - The two statements refer to different epidemiological categories of Candida infections. The estimated ~700,000 cases of invasive candidiasis per year correspond specifically to systemic and bloodstream infections (candidemia and deep-seated invasive infections), which represent the most severe and life-threatening forms of disease caused by Candida spp. In contrast, the broader estimate of ~13 million fungal infections and 1.5 million deaths annually refers to all major fungal diseases worldwide, with Candida spp., particularly Candida auris, being among the leading causative agents in hospital and intensive care settings. Therefore, the two figures are not contradictory but rather describe different epidemiological levels — one specific to invasive candidiasis, and the other encompassing the global burden of fungal infections in general.

directly after the author et al. Edit the entire manuscript. - Thank you for this valuable comment; the entire text has been revised accordingly.

‘Carbon dots can also penetrate Candida biofilms, deliver antifungal agents, and generate

reactive oxygen species while remaining biocompatible’ Add citation - Citations included in Table 1.

‘Chitosan-based nanoparticles disrupt 264 fungal membranes and inhibit adhesion’ Add

Citation – Citations included in Table 1.

‘Liposomes have long been used to encapsulate antifungal agents, improving their

penetration and reducing toxicity. Related lipid-based carriers, including solid lipid

nanoparticles (SLNs) and nanocapsules, provide controlled release and stability

advantages. Enzyme-mimicking nanoenzymes further contribute by catalytically degrading

extracellular biofilm components and producing fungicidal radicals. Table 1 provides an

overview of the key advantages and limitations of the aforementioned nanomaterials, as

well as their potential applications in photocatalytic strategies against Candida biofilms’

Add citations - Citations included in Table 1.

‘Table 1’ the column ‘Applied in photocatalysis against biofilm Candida spp. – subsection’.

It is not clear to the reviewer. What did the author mean? - Thank you for pointing out this issue. The intention of the column entitled “Applied in photocatalysis against biofilm Candida spp. – subsection” was to indicate whether, according to the current state of knowledge, the specific type of nanomaterial has been reported or recognized as applicable in photocatalytic processes for the inactivation or eradication of Candida biofilms.

‘extracellular polymeric substance (EPS)’ After the first time, it is sufficient to report EPS;

the same applies to the ROS, the hydroxyl radical, etc. Check the whole manuscript.  Thank you for this important comment; the text has been corrected.

‘Among the tested microorganisms’. Specify the microorganisms. – Done.

Once again, we would like to express our sincere gratitude for your valuable review.

Thank you in advance!

Regards,

prof. Emil Paluch

Reviewer 3 Report

Comments and Suggestions for Authors

The manuscript by Karolina Kraus et al. entitled ‘Nanomaterials for Photocatalytic Inactivation and Eradication of Candida spp. Biofilms in Healthcare Environment: A Novel Approach in Modern Clinical Practice’ aims to summarize the knowledge on the potential use of nanomaterials exhibiting photodynamic properties as a tool for combating Candida biofilms. The topic is timely, as Candida-induced infections are on the rise, along with the increase in the drug-resistance of Candida.

The authors structured the manuscript nicely, the Figures illustrate the knowledge being conveyed quite vividly. The manuscript is well written and fits the scope of the journal Molecules.

However, I recognized some issues associated with the review:

  1. My biggest complaint would be how broad the subject of the manuscript is. At times I was not sure whether the authors wanted to describe the anti-biofilm antimicrobial PDT of Candida or simply the development of the materials suitable for surface disinfection. These two subjects are quite distinctive and are not interchangeable in terms of the materials activity and desired features. The manuscript should be limited to one potential application, either for administration in patients or for medical devices. Secondly, the sheer number of photoactive nanomaterials seems to be too much to enable each group characterization within two or three paragraphs. As an example, this results in referencing only three papers in 5.1. section while the authors mention themselves that TiO2 is the most researched photocatalysts (‘TiOâ‚‚ being among the most commonly used nanoparticles with catalytic properties’ – lines 904-905). Because of all these reasons, it seems that the article lacks depth in the matter of various nanomaterials tested (both the chemical composition and structural differences).
  2. The studies cited seem to be chosen at random – it is difficult to draw any conclusion based on the information that is provided in the review, except that these nanomaterials act photocatalytically on Candida biofilms.
  3. The manuscript also lacks some of the drawbacks of potential application of nanoparticles –i.e., light absorption spectrum, scale-up, stability, etc.
  4. Two sections are numbered with 3 (3. Candida spp. biofilm resistance factors; 3. Nanomaterials against Candida spp. biofilms).

Section 2 (‘Biofilm formation by Candida spp.’) is in my opinion the best written part of the paper.

Author Response

We sincerely thank you for the thorough and constructive review of our manuscript, as well as for all the positive observations. We are pleased that Section 2 was found to be well-structured. Your comments have provided us with the opportunity to re-examine and improve our work, for which we are very grateful. We would like to address each of your suggestions in detail below:

  1. My biggest complaint would be how broad the subject of the manuscript is. At times I was not sure whether the authors wanted to describe the anti-biofilm antimicrobial PDT of Candida or simply the development of the materials suitable for surface disinfection. These two subjects are quite distinctive and are not interchangeable in terms of the materials activity and desired features. The manuscript should be limited to one potential application, either for administration in patients or for medical devices. Secondly, the sheer number of photoactive nanomaterials seems to be too much to enable each group characterization within two or three paragraphs. As an example, this results in referencing only three papers in 5.1. section while the authors mention themselves that TiO2 is the most researched photocatalysts (‘TiOâ‚‚ being among the most commonly used nanoparticles with catalytic properties’ – lines 904-905). Because of all these reasons, it seems that the article lacks depth in the matter of various nanomaterials tested (both the chemical composition and structural differences). – In writing our manuscript, we intended it to serve as a narrative review, aiming to highlight modern, rather than widely established, approaches to the use of photocatalysis for combating Candida biofilms in healthcare environments. Indeed, the topic is broad and multifaceted; however, we believe that this breadth allows us to compile, in one place, the most important practical insights for readers interested in healthcare applications, who wish to explore and implement the described strategies in clinical practice. For this reason, we deliberately chose not to expand on topics that are already widely established, such as the use of TiOâ‚‚. The diversity of topics addressed in our review was meant to draw attention to the range of solutions whose potential has not yet been fully realized in practice, and to demonstrate the broad possibilities that photocatalysis offers for practical medicine. After receiving all the reviews, we have refined certain aspects of the manuscript to ensure that it is as clear and accessible as possible.
  2. The studies cited seem to be chosen at random – it is difficult to draw any conclusion based on the information that is provided in the review, except that these nanomaterials act photocatalytically on Candida biofilms. – Due to the limited prevalence of research studies examining the methods we describe, it was not possible to compare the effects of the individual strategies directly. We hope that the topic will be further explored in the future, allowing such analyses to be conducted, which we would be eager to undertake. Our aim was to select valuable novel studies that demonstrated promising results, which should be further validated in larger-scale research. This could potentially enable broader application of the described methods in clinical practice. The topic appears to be not yet widely explored, which motivated us to prepare a review providing a comprehensive overview and a set of suggestions for how these approaches could be further developed and implemented in practice.
  3. The manuscript also lacks some of the drawbacks of potential application of nanoparticles –i.e., light absorption spectrum, scale-up, stability, etc. – We thank you for this important comment. Indeed, our manuscript does not include such a comparison, which was intentional, as we aimed to provide a concise overview rather than replicate information that is already extensively described in other sources focusing specifically on nanomaterials and their properties, rather than their healthcare practical usage. In line with the objective of our study, we instead included a table summarizing the fundamental advantages and disadvantages of the materials discussed within the context of healthcare.
  4. Two sections are numbered with 3 (3. Candida spp. biofilm resistance factors; 3. Nanomaterials against Candida spp. biofilms). – We have corrected the numbering.

Once again, we thank you for your review and hope that the vision and objectives of our work are clear and convincing.

Thank you in advance!

Regards,

prof. Emil Paluch

Reviewer 4 Report

Comments and Suggestions for Authors

Dear Authors,

Your paper has great potential, illustrating a significant challenge in dealing with infections caused by Candida spp. and the biofilms they form.

In your work, you showed that the use of nanomaterials, such as nanoparticles, carbon dots, or nanozymes, for photocatalytic processes seems to be a promising solution, showing outstanding results in Candida spp. biofilm disruption and inactivation.

Author Response

We would like to sincerely thank you for your very positive evaluation and encouraging feedback on our manuscript.
We truly appreciate your recognition of the scientific relevance and potential of our work.
Your supportive comments are very motivating for our research team, and we are grateful for your time and effort in reviewing our paper.

Thank you in advance!

Regards,

prof. Emil Paluch

Round 2

Reviewer 1 Report

Comments and Suggestions for Authors

The authors have made careful revisions and the manuscript can be acceptable now.

Reviewer 3 Report

Comments and Suggestions for Authors

The provided changes improve the manuscript significantly. The authors' response clarifies the issues raised in my previous review.